# Lyra: Orchestrating Dual Correction in Automated Theorem Proving

## Abstract

Large Language Models (LLMs) present an intriguing avenue for exploration in the field of formal theorem proving. Nevertheless, their full potential, particularly concerning the mitigation of hallucinations and refinement through prover error messages, remains an area that has yet to be thoroughly investigated. To enhance the effectiveness of LLMs in the field, we introduce the Lyra, a new framework that employs two distinct correction mechanisms: *Tool Correction* (TC) and *Conjecture Correction* (CC). To implement *Tool Correction* in the post-processing of formal proofs, we leverage prior knowledge to utilize predefined prover tools (e.g., Sledgehammer) for guiding the replacement of incorrect tools. *Tool Correction* significantly contributes to mitigating hallucinations, thereby improving the overall accuracy of the proof. In addition, we introduce *Conjecture Correction*, an error feedback mechanism designed to interact with prover to refine formal proof conjectures with prover error messages. Compared to the previous refinement framework, the proposed *Conjecture Correction* refines generation with instruction but does not collect paired (generation, error & refinement) prompts. Our method has achieved state-of-the-art (SOTA) performance on both miniF2F validation ($48.0\% \rightarrow 55.3\%$) and test ($45.5\% \rightarrow 51.2\%$). We also present 3 IMO problems solved by Lyra. We believe *Tool Correction* (post-process for hallucination mitigation) and *Conjecture Correction* (subgoal adjustment from interaction with environment) could provide a promising avenue for future research in this field.

## 1 Introduction

Formal proof automation is a challenging task that has garnered increased attention in recent years (Bansal et al., 2019a; Polu & Sutskever, 2020; Lample et al., 2022; Jiang et al., 2022; Wu et al., 2022; Wang et al., 2023b). Unlike other domains where deep learning approaches have shown remarkable success, previous studies have proposed techniques to synthesize additional formal training data (Wu et al., 2022; Polu & Sutskever, 2020; Han et al., 2021; Bansal et al., 2019b; Polu et al., 2023). Recently, large language models (LLMs) trained on informal mathematical data have showcased impressive quantitative reasoning abilities (Lewkowycz et al., 2022; Welleck et al., 2022).

Draft, Sketch, and Prove (DSP) (Jiang et al., 2023) maps informal proofs to formal proof sketches, and uses the sketches to guide an automated prover by directing its search to easier sub-problems. Following this direction, Subgoal-based Learning (Zhao et al., 2023) replaces the informal proof with subgoal-proof and learns how to optimize subgoal demonstration selection. However, they have not been able to post-process LLM generation or gradually refine previous generations.

In this paper, we seek to build Lyra based on LLM, focusing on formal theorem proving. There are two major challenges for LLM generation: 1) hallucination mitigation; 2) interaction with the environment. To mitigate LLM hallucination, we propose *Tool Correction* to leverage prior knowledge and rules to guide incorrect tool replacement. As shown in the observation in Figure 1, prover fails to prove conjecture $x = 19 * (x\ div\ 19) + 4$ because by (simp add: div_mult_mod_eq) generated by LLM cannot prove $x = 19 * (x\ div\ 19) + 4$ (considered as LLM hallucination), while the conjecture is correct but employed tool simp is not powerful enough. *Tool Correction* employs predefined tools (e.g. sledgehammer, arith) to guide incorrect tool replacement and finally prove the conjecture. We also propose a general interaction technique with LLM named *Conjecture Correction*. To further improve and modify the conjectures, *Conjecture Correction* leverages a general framework

Figure 1: **Our proposed Lyra framework contains two modules**. *Tool Correction*: employ the predefined tools to replace the incorrect tools and prove the conjectures. The prover fails because by (simp add: div_mult_mod_eq) generated by LLM cannot prove $x = 19 * (x \ div \ 19) + 4$, which is considered as the hallucination. Actually, the conjecture is correct and simple, and the prover fails to prove it because it employs an incorrect tool. Hence, the prover successfully proves the conjecture when employing by arith. *Conjecture Correction*: We design an interaction framework that integrates previous formal sketch and prover error messages to better sketch generation. The steps with the ATPWithTC delimiters are generated by an automated prover with *Tool Correction*.

that can easily integrate feedback from any environment, in this case, the Isabelle prover, to further polish conjectures. We believe the Lyra presents our insights to mitigate LLM hallucination and interact with the environment.

The proposed method significantly outperforms competing approaches in formal theorem-proving tasks, achieving a pass rate of $51.2\%$ on the miniF2F test dataset, a $5.7\%$ absolute improvement over the previous state-of-the-art. Furthermore, the insights gained from *Tool Correction* and *Conjecture Correction* design can be applied to other frameworks that need to interact with the environment. In summary, our contributions are as follows:

- We introduce Lyra, a method composed of two components *Tool Correction* and *Conjecture Correction*, to guide automated provers with formal proof sketches.

- *Tool Correction* employs the predefined tools to replace the incorrect tools to mitigate hallucination, while *Conjecture Correction* integrates previous formal sketch and prover error messages to refine proof.

- We establish a new SOTA of 55.3% and 51.2% on miniF2F validation and test, outperform previous best 7.3% and 5.7% respectively. And we newly solve two IMO problems: `IMO_1974_p5` and `IMO_1981_p6`.

## 2 RELATED WORKS

**Interactive theorem provers.** Contemporary mathematical verification systems are centered on interactive theorem provers (ITPs), including Isabelle (Paulson, 1994), Lean (de Moura et al., 2015), Coq (Barras et al., 1997), and Metamath (Megill & Wheeler, 2019). ITPs lay the groundwork for mathematical definitions and theorems on a robust logical foundation through their core kernels. The validation of each theorem is kernel-based and takes place within the ITP. To achieve formal proof, a theorem is initially expressed in the programming language of the ITP and systematically refined into simpler subgoals until it aligns with previously established facts. In this paper, the chosen ATP is Isabelle, known for its potent prover tools, including sledgehammer (Paulson, 2010).

**Machine learning for formal proving.** Numerous approaches advocate the integration of machine learning with contemporary interactive theorem provers (ITPs) (Yang & Deng, 2019; Gauthier et al., 2021). They leverage the recent advancements in language models (Polu & Sutskever, 2020; Han et al., 2021; Polu et al., 2023; Jiang et al., 2022; Lample et al., 2022; Mikuła et al., 2023). These techniques recommend actions based on the current proving state, and the tree search identifies a sequence of correct steps using actions provided by the language model. Potent methods like MCTS (Silver et al., 2018; Wu et al., 2021; Laurent & Platzer, 2022) or dynamic-tree MCTS (Wang et al., 2023b) are utilized for this purpose. Previous work (Wu et al., 2022) has demonstrated the few-shot statement autoformalization capability of LLMs (Chowdhery et al., 2022). In investigating these findings' applicability to proof autoformalization, DSP (Jiang et al., 2023) conducted an in-depth analysis using Draft, Sketch, and Proof. Subgoal-Learning (Zhao et al., 2023) further employs a subgoal-goal-based informal proof approach. In an effort to support the open-source community, LeanDojo (Yang et al., 2023) created a Lean playground that includes toolkits, data, models, and benchmarks. While these methods directly use the results generated by LLMs, we adopt a different approach by employing predefined tools to post-process the generations to mitigate hallucination, specifically *Tool Correction*.

**Large language model refinement.** Calibration studies conducted on LLMs reveal that the probabilistic predictions made by current LLMs are closely aligned with the actual frequencies of token occurrences, resulting in well-calibrated predictions for specific tasks (Guo et al., 2017; Kadavath et al., 2022; Jiang et al., 2020). As LLMs exhibit reliable calibration, an increasing number of research studies emphasize using self-evaluation for verification. For instance, Reflexion (Shinn et al., 2023) leverages an agent with dynamic memory and self-reflection capabilities, while Self-Refine (Madaan et al., 2023) proposes a method to generate outputs from LLMs and refine their previously generated outputs based on their own feedback. Taking a similar approach, methods like Self-Debug (Chen et al., 2023) and CRITICS (Gou et al., 2023) interact with code interpreters to further debug. The Baldur (First et al., 2023) trains a model to correct and refine the generated proof. In contrast, Progressive-Hint Prompting (Zheng et al., 2023) iteratively extracts hints from previous LLM's answers as hints for the next answer generation. However, previous works require extensive prompts, including generation prompts and refine prompts. Our approach *Conjecture Correction* refines generation with instruction but does not collect paired (generation, error & refinement) prompts.

## 3 METHOD

This section describes our Lyra for formal proof automation, which leverages *Tool Correction* and *Conjecture Correction* to guide automated formal theorem provers with proof sketches.

### 3.1 BACKGROUND: PIPELINE OF DSP

DSP (Jiang et al., 2023) aims to generate a formal sketch from an informal statement, verifiable by an off-the-shelf automated theorem prover. DSP creates $N$ demonstration examples, denoted as $E = E_1, E_2, ..., E_N$, each containing informal/formal components (statements, proofs, sketches). The pipeline of DSP has the following three steps.

---

**Algorithm 1** Pseudocode of *Tool Correction* in a Python-like style.

---

```python
#tactic_list: list of the tactics of formal proof
#prover: Isabelle Prover
#TCUsage: whether employ Tool Correction
tool_heuristics=['by auto','by arith','by blast', 'by simp',
'by fastforce', 'by force', 'by eval', 'by presburger', 'by sos',
'by linarith', 'by (auto simp: field_simps)', 'sledgehammer']
for tactic in tactic_list:
    use_heuristics=False
    output = prover.run_tac(tactic)
    if output['error'] is not None:
        if TCUsage: # Use Tool Correction or Not
            if tactic.strip().startswith("by") or tactic.strip()==("."):
                use_heuristic=True

        if ("sledgehammer" in tactic) or use_heuristic:
            for tool_try in tool_heuristics:
                output = prover.run_tac(tool_try)
                if output['error'] is None:
                    break
    if output['error'] is not None:
        return "tactic_failed", output
    if output['tactic_state'] == 'no goals':
        return "success", output

return "proof_incomplete", output
```

---

**Informal proof generation.** There are two scenarios: one with an existing human informal proof and another where a language model generates draft-proof candidates without a human reference. For LLM-generated informal proof, DSP provides the model with a few examples containing both (statement, informal proof) for informal proof generation. Subsequently, DSP presents a problem statement that needs to be translated and the model then generates the subsequent tokens to produce the desired informal proof.

**Formal proof generation.** DSP leverages the few-shot learning capabilities of a large language model. Specifically, DSP provides the model with a few example pairs containing (statement, informal proof, formal sketch) for formal proof generation. Subsequently, DSP presents a (statement, informal proof) that needs to be translated. The model then generates the subsequent tokens to produce the desired formal sketch.

**Prover validation.** In the final phase, off-the-shelf automated provers address sketch gaps. These systems create formally valid proofs. DSP framework remains agnostic to prover type (symbolic, neural, hybrid). Successful prover results yield verifiable formal proofs.

## 3.2 TOOL CORRECTION

*Tool Correction* employs prior knowledge to employ predefined tools (e.g. sledgehammer) to guide incorrect tool replacement, as shown in Algorithm 1. We introduce the *Tool Correction* as a remedy to alleviate the generation errors stemming from Large Language Models (LLMs). Through empirical observation, it becomes evident that despite the factual accuracy of conjectures, LLMs at times adopt misguided tools that do not withstand validation by theorem provers, as shown in Figure 1.

For instance, consider the statement $x = 19 * (x \ div \ 19) + 4$, where LLM proposes to utilize the tactic by (simp add: div_mult_mod_eq), leading to failure. This is the LLM hallucination, as by (simp add: div_mult_mod_eq) is suited for proving $a = a \ div \ b * b + a \ mod \ b$ but not $x = 19 * (x \ div \ 19) + 4$. Substituting it with by arith enables the theorem prover to successfully verify $x = 19 * (x \ div \ 19) + 4$. Hence, in certain instances, LLM might formulate correct conjectures but employ inappropriate tools, resulting in unsuccessful proof attempts. To address this, *Tool Correction* leverages predefined tools to enhance the success rate.

---

**Algorithm 2** Pseudocode of *Conjecture Correction* in a Python-like style.

---

```python
#round_count: the current round number
#prompt_sample: the prompt and proposed question
#previous_response: previous formal proof
#error_info: error information from Isabelle
input=[{"role": "system",  "content": "You are an expert in \
Mathematical Proof and Isabelle Proof Assistant. Follow the given \
examples and complete the proof with Isabelle Proof Assistant"},
{"role": "user", "content": prompt_sample}]
if round_count%5!=0: #If Falase, the initial round.
#Otherwise, then the Refinement round.
 #Refinement Rround
  input.append({"role": "assistant", "content": previous_respon})
  input.append({"role": "user", "content": "(*The last proof has the \
  following errors from Isabelle Prover. Therefore,\n 1) Please Follow  \
  the Above Prompt;\n\n 2) And Utilize the Following Errors to redo \
  the last formal proof.\n {}.\n\n*)\n\n \
  proof -\n".format(error_info)})
json_obj = openai.ChatCompletion.create(messages=input)
```

---

The *Tool Correction* approach entails the validation of a given tactic $t$ using Isabelle. If validation succeeds, we proceed; if not, *Tool Correction* intervenes to modify the tactic. Specifically, when a tactic is equal to "." or commencing with "by" or "sledgehammer" but the tactic fails, we attempt the application of $t_{tool}$. This $t_{tool}$ can be either: 1) "sledgehammer" or; 2) by `tool` with `tool` belonging to the set (`auto`, `simp`, `blast`, `fastforce`, `force`, `eval`, `presburger`, `sos`, `arith`, `linarith`, `auto simp:  field simps`).

By integrating *Tool Correction*, we systematically explore the applicability of "sledgehammer" and 11 heuristic tools. If any of these successfully pass the theorem prover, we progress to the subsequent tactics. However, if proof still fails to prove the tactic after trying all $t_{tool}$ fail, the overall proof attempt is deemed unsuccessful.

## 3.3 CONJECTURE CORRECTION

For *Conjecture Correction*, we design a framework that can easily integrate previous formal sketches and error messages from the prover to improve sketch generation. LLMs, particularly GPT-4, can leverage prior responses or contextual cues for improved output. Nonetheless, integrating feedback into mathematical proving remains a challenge. This stems from two primary factors: 1) diverse theorem provers employ distinct syntax, complicating the design of varied prompts; 2) often require an extensive token count, incurring a high computational cost and exceeding model length limits. To address these limitations, Lyra uses *Conjecture Correction*, offering a versatile refinement pipeline that can transform a non-refined framework into a refined one. Compared to the previous refinement framework, such as Self-Refine (Madaan et al., 2023) or Self-Debug (Chen et al., 2023), the proposed *Conjecture Correction* refines generation with instruction, but does not collect paired (generation, error & refinement) prompts. The details are shown in Algorithm 2.

**Initial round generation.** In the initial round generation, we follow the same process as DSP, directly producing informal or formal proofs without prover error messages.

**Rectification round.** Our approach also involves the use of an LLM (e.g. GPT-4 (Bubeck et al., 2023)) for rectification. In contrast to the initial round generation, the rectification employs the same initial prompt as the first round but appends error messages from the prover. As most formal proofs of our prompts begin with `proof -`, we add `proof -` at the end of the instruction so that the LLM response is formal proof.

**Reset initial round generation.** Since the rectification round builds upon the first round, the quality of *Conjecture Correction* performance is tied to the initial round proof. To ensure that a potentially subpar initial round proof does not negatively affect subsequent proofs, we regenerate the initial round proof at interaction rounds $K$, $2K$, $3K$ and so on, refining its generation in the remaining rounds. For example, when working with 200 attempts and setting $K$ to 5, *Conjecture Correction*

Table 1: **Proving success rates on the miniF2F dataset with Isabelle.** The table displays the success rates of previous works and the Lyra, using both human and GPT-4 informal proofs. The highest success rates for each set are highlighted in bold.

| Success rate | miniF2F-valid | miniF2F-test |
|---|---|---|
| *Baselines* | | |
| Sledgehammer (Paulson, 2010) | 9.9% | 10.4% |
| Sledgehammer + heuristics (Jiang et al., 2023) | 18.0% | 20.9% |
| Thor (Jiang et al., 2022) | 28.3% | 29.9% |
| Thor + expert iteration (Wu et al., 2022) | 37.3% | 35.2% |
| *Draft, Sketch, and Prove (100 attempts)* (Jiang et al., 2023) | | |
| Human informal proof | 42.6% | 39.3% |
| 540B Minerva informal proof | 42.6% | 38.9% |
| *Subgoal-Learning (100 attempts)* (Zhao et al., 2023) | 48.0% | 45.5% |
| *Lyra (Ours)* | | |
| GPT-4 informal proof (100 attempts) | 52.8% | 44.2% |
| GPT-4 informal proof (200 attempts) | 54.9% | 47.9% |
| Human informal proof (100 attempts) | 52.0% | 47.1% |
| Human informal proof (200 attempts) | **55.3**% | **51.2**% |

partitions the 200 attempts into 40 patches. Each patch consists of the first proof derived from DSP, followed by four subsequent refined proofs that build upon the previous proof and incorporate the error message provided by the prover.

## 4 EXPERIMENT

### 4.1 DATASET

In this study, we assess our approach using the miniF2F dataset (Zheng et al., 2021), which is a collection of 488 formal mathematical problems derived from high-school competitions and expressed in three formal languages: Lean (de Moura et al., 2015), HOL-Light (Bansal et al., 2019a), and Isabelle (Paulson, 1994). The dataset is divided into validation and test sets, each containing 244 problems. These problems are sourced from three distinct categories, with 260 problems extracted from the MATH dataset (Hendrycks et al., 2021), 160 problems taken from actual high school mathematical competitions (AMC, AIME, and IMO), and 68 problems specially crafted to mirror the difficulty level of the aforementioned competitions.

**Evaluation.** The objective of our study is to generate formal sketches for the problems in the miniF2F dataset. We consider a proof valid if and only if (a) it does not have any "cheating" keywords (`sorry` and `oops`) that terminate a proof without completion, and (b) Isabelle must be capable of verifying the corresponding formal statement with the proof.

**Implementation details.** In our research, we utilized GPT-4 as the Language Model Model (LLM) for generating informal drafts and formal sketches. The temperature of GPT-4 was set to 0.7, with 200 attempts. The details of baselines are shown in Appendix.

### 4.2 MAIN RESULTS

Table 1 presents the distribution of successful formal proofs obtained from the miniF2F dataset using the interactive theorem prover Isabelle. An examination of the results presented in Table 1 reveals a conspicuous enhancement in the efficacy of the Sledgehammer automated prover, owing to the integration of 11 supplementary heuristic tactics (Jiang et al., 2023). Noteworthy achievements are also realized through deploying the DSP-based methods (DSP and Subgoal), attaining success rates of 39.3% and 45.5%, respectively on the miniF2F test set.

By harnessing informal proofs generated by GPT-4, our proposed method achieves success rates of 54.9% and 47.9% on the validation and test sets of miniF2F respectively. This performance persists

Table 2: **Ablation results on the miniF2F dataset with Isabelle.** There are three important conclusions: 1) GPT-4 is better than Codex for mathematical proving; 2) *Tool Correction* can consistently improve performance; 3) *Conjecture Correction* can improve performance but needs more attempts. **Our proposed method degrades to DSP (**Jiang et al., **2023) when without** *Tool Correction* **and** *Conjecture Correction*.

| Attempt | Formal Proof | Informal Proof | TC | CC | miniF2F-valid | miniF2F-test |
|---|---|---|---|---|---|---|
| 100 | Codex | 540B Minerva | ✗ | ✗ | 42.6% | 38.9% |
| | GPT-4 | GPT-4 | ✗ | ✗ | 48.3% | 38.9% |
| | Codex | Human | ✗ | ✗ | 42.6% | 39.3% |
| | GPT-4 | Human | ✗ | ✗ | 47.9% | 39.7% |
| | GPT-4 | GPT-4 | ✓ | ✓ | 52.8% | 44.2% |
| | GPT-4 | Human | ✓ | ✓ | 52.0% | 47.1% |
| 200 | GPT-4 | GPT-4 | ✗ | ✗ | 49.5% | 40.9% |
| | GPT-4 | GPT-4 | ✓ | ✗ | 55.3% | 45.0% |
| | GPT-4 | GPT-4 | ✗ | ✓ | 48.3% | 40.9% |
| | GPT-4 | GPT-4 | ✓ | ✓ | 54.9% | 47.9% |
| | GPT-4 | Human | ✗ | ✗ | 50.4% | 42.6% |
| | GPT-4 | Human | ✓ | ✗ | 52.8% | 45.9% |
| | GPT-4 | Human | ✗ | ✓ | 46.7% | 43.0% |
| | GPT-4 | Human | ✓ | ✓ | **55.3**% | **51.2**% |

even when the attempt number is set at 100, affirming its robustness. When the attempt number is 100, compared to 540B Minerva informal proof with DSP, our proposed Lyra improves the performance on miniF2F validation set from 42.6% to 52.8% and miniF2F test set from 38.9% to 44.2%. This outcome can be attributed to the *Tool Correction* and *Conjecture Correction*.

In instances where human informal proofs are employed, our proposed method demonstrates impressive success rates of 55.3% and 51.2% on the validation and test sets of miniF2F. Comparative analysis against DSP reveals an improvement of 12.7% and 11.9% on the validation and test sets respectively for miniF2F. Furthermore, when contrasted with the previous state-of-the-art Subgoal-Learning model, our approach showcases an advancement of 7.3% and 5.7% on the miniF2F validation and test sets respectively.

The performance of human informal proofs surpasses that of GPT-4 generated counterparts, especially on the test set. This substantiates the notion that precision in informal proofs is important for generating formal sketches.

## 4.3 ABLATION STUDY

**GPT-4 is better than Codex, especially on miniF2F validation dataset..** In the absence of *Tool Correction* and *Conjecture Correction*, our proposed method experiences degradation to DSP. Referring to Table 2, when considering the informal proof generated by LLM (GPT-4 or 540B Minerva), GPT-4 is better than Codex (Chen et al., 2021). When compared with the deployment of Codex for generating formal sketches, GPT-4 demonstrates improvements of 5.3% and 0.4% on the validation and test subsets of miniF2F, respectively, while utilizing the same attempt number 100 and human informal proof. This substantiates the notion that GPT-4 indeed enhances performance.

***Tool Correction*: consistently improve performance.** As evident from Table 2 and Figure 2, the inclusion of *Tool Correction* yields enhanced performance. Similarly, when assessing GPT-4-generated informal proofs on the miniF2F test set, *Tool Correction* elicits improvements of 4.1% and 7.0% in the absence and presence of *Conjecture Correction*, respectively. When considering human informal proofs on the miniF2F test set, *Tool Correction* showcases enhancements of 3.3% and 8.2% in scenarios devoid of and accompanied by *Conjecture Correction*, respectively. Therefore, regardless of whether the informal sketch is generated by GPT-4 or created manually by a human, *Tool Correction* consistently enhances performance and can further benefit from the addition of *Conjecture Correction*.

***Conjecture Correction*: further improves performance, prefers more powerful prover and requires more attempts to be convergent.** The outcomes presented in Table 2 and illustrated

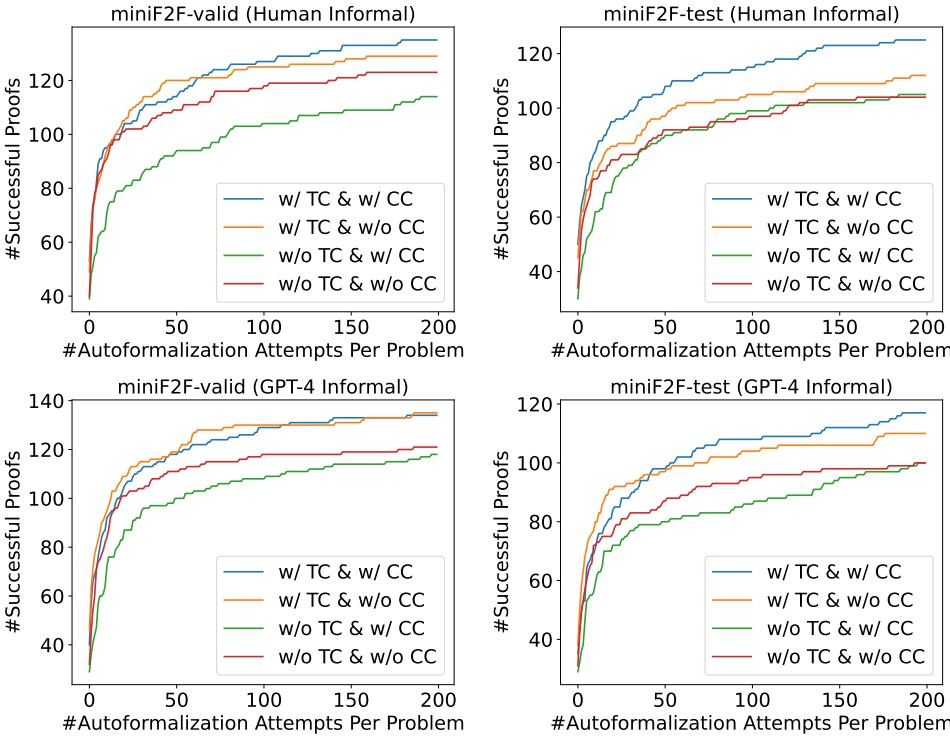

Figure 2: **Number of problems solved on miniF2F against the number of autoformalization attempts per problem**. On miniF2F validation and test set, we have shown the results of *Tool Correction* (TC) and *Conjecture Correction* (CC) on human informal proof and GPT-4 informal proof respectively.

in Figure 2 underscore the efficacy of integrating *Conjecture Correction*, albeit at the expense of requiring an increased number of attempts to achieve convergence. When considering human informal proofs on the miniF2F test set, *Conjecture Correction* showcases enhancements of 0.4% and 5.3% in scenarios devoid of and accompanied by *Tool Correction*, respectively. This suggests that *Conjecture Correction* improves proof quality, but needs a more powerful prover (e.g. with *Tool Correction*) to fill the formal gaps. *Conjecture Correction* needs more attempts to be convergent because *Conjecture Correction* modifies the initial proof to generate subsequent proofs, which strongly hinges on the quality of the initial proof. Specifically, *Conjecture Correction* partitions the pool of 200 attempts into 40 patches, wherein the first proof originates from DSP, and the subsequent four are based on the initial proof. Furthermore, it's worth noting that, in theory, any problems solvable through DSP remain solvable using our approach, as DSP is equivalent to our initial proof generation without *Tool Correction*.

**Attempt number: Lyra benefits more with attempt number increment.** In the absence of *Tool Correction* and *Conjecture Correction*, our proposed method reduces to DSP. Within the validation set with human informal proofs, when the number of attempts is escalated from 100 to 200 (shown in Table 2), the performance of DSP experiences a gain from 47.9% to 50.4%, achieving a 2.5% improvement. Conversely, our proposed approach exhibits a performance improvement from 52.0% to 55.3%, reflecting a more substantial 3.3% enhancement. For the test set, DSP's performance improves from 39.7% to 42.6%, marking a 2.9% increment. In contrast, our method demonstrates an increment from 47.1% to 51.2%, indicating a more 4.1% boost. This divergence implies that our proposed approach effectively surpasses the performance limitations of DSP, highlighting the potential efficacy of expanding the attempt number to further enhance performance differences.

## 4.4 CASE STUDY

We solve another IMO problem `IMO_1959_p1` with GPT-4 informal proof, which is also solved via DSP with 540B Minerva. Furthermore, to present the effectiveness of our method, we provide a formal sketch of an IMO problem named `IMO_1974_p5` that remains unproven by earlier state-of-the-art methods. As demonstrated in Figure 3, our Lyra successfully proves `IMO_1974_p5` with

**Statement:** Determine all possible values of $S = \frac{a}{a+b+d} + \frac{b}{a+b+c} + \frac{c}{b+c+d} + \frac{d}{a+c+d}$ where $a$, $b$, $c$, $d$, are arbitrary positive numbers.

**Informal Proof (Human):**
Note that $2 = \frac{a}{a+b} + \frac{b}{a+b} + \frac{c}{c+d} + \frac{d}{c+d} > S > \frac{a}{a+b+c+d} + \frac{b}{a+b+c+d} + \frac{c}{a+b+c+d} + \frac{d}{a+b+c+d} = 1$. We will now prove that $S$ can reach any range in between 1 and 2.
Choose any positive number $a$. For some variables such that $k, m, l > 0$ and $k + m + l = 1$, let $b = ak$, $c = am$, and $d = al$. Plugging this back into the original fraction, we get $S = \frac{a}{a+ak+al} + \frac{ak}{a+ak+am} + \frac{am}{ak+am+al} + \frac{al}{a+am+al} = \frac{1}{1+k+l} + \frac{k}{1+k+m} + \frac{m}{k+m+l} + \frac{l}{1+m+l}$. The above equation can be further simplified to $S = \frac{1}{2-m} + \frac{k}{2-l} + m + \frac{l}{2-k}$. Note that $S$ is a continuous function and that $f(m) = m + \frac{1}{2-m}$ is a strictly increasing function. We can now decrease $k$ and $l$ to make $m$ tend arbitrarily close to 1. We see $\lim_{m \to 1} m + \frac{1}{2-m} = 2$, meaning $S$ can be brought arbitrarily close to 2. Now, set $a = d = x$ and $b = c = y$ for some positive real numbers $x, y$. Then $S = \frac{2x}{2x+y} + \frac{2y}{2y+x} = \frac{2y^2+8xy+2x^2}{2y^2+5xy+2x^2}$. Notice that if we treat the numerator and denominator each as a quadratic in $y$, we will get $1 + \frac{g(x)}{2y^2+5xy+2x^2}$, where $g(x)$ has a degree lower than 2. This means taking $\lim_{y \to \infty} 1 + \frac{g(x)}{2y^2+5xy+2x^2} = 1$, which means $S$ can be brought arbitrarily close to 1. Therefore, we are done.

**Formal Proof:**

```
theorem imo_1974_p5:
  fixes a b c d s :: real
  assumes "a>0" "b>0" "c>0" "d>0"
  assumes h0 : "s=a/(a+b+d) + b/(a+b+c) + c/(b+c+d) + d/(a+c+d)"
  shows "1<s \<and> s<2"
proof -
(* Let's correct the proof. *)
  (* Define the sum S *)
  let ?S = "a/(a+b+d) + b/(a+b+c) + c/(b+c+d) + d/(a+c+d)"
  (* Show that S > 1 *)
  have "a/(a+b+c+d) + b/(a+b+c+d) + c/(a+b+c+d) + d/(a+b+c+d) = 1"
    using assms by (simp add: divide_simps) [ATPWithTC]
  moreover have "?S > a/(a+b+c+d) + b/(a+b+c+d) + c/(a+b+c+d) + d/(a+b+c+d)"
  proof -
    have "a/(a+b+d) > a/(a+b+c+d)" "b/(a+b+c) > b/(a+b+c+d)"
         "c/(b+c+d) > c/(a+b+c+d)" "d/(a+c+d) > d/(a+b+c+d)"
      using assms by (simp_all add: divide_strict_right_mono) [ATPWithTC]
    then show ?thesis  by auto [ATPWithTC]
  qed
  ultimately have "1 < ?S" by simp [ATPWithTC]
  (* Show that S < 2 *)
  have "a/(a+b) + b/(a+b) + c/(c+d) + d/(c+d) = 2"
    using assms  by (simp add: divide_simps) [ATPWithTC]
  moreover have "?S < a/(a+b) + b/(a+b) + c/(c+d) + d/(c+d)"
  proof -
    have "a/(a+b+d) < a/(a+b)" "b/(a+b+c) < b/(a+b)"
         "c/(b+c+d) < c/(c+d)" "d/(a+c+d) < d/(c+d)"
      using assms by (simp_all add: divide_strict_right_mono)  [ATPWithTC]
    then show ?thesis  by auto [ATPWithTC]
  qed
  ultimately have "?S < 2" by simp [ATPWithTC]
  (* Conclude the proof *)
  then show "1<s \<and> s<2" using assms h0 `1 < ?S`  by auto [ATPWithTC]
qed
end
```

Figure 3: **A successful formal proof synthesized with human informal proof**. With *Tool Correction* and *Conjecture Correction*, we successfully solve an IMO problem `IMO_1974_p5`. The steps with the ATPWithTC delimiters are generated by an automated prover with *Tool Correction*. We also solve `IMO_1959_p1` with GPT-4 informal proof, which is shown in the Appendix.

*Tool Correction* and *Conjecture Correction*. We have shown the interaction details of `IMO_1974_p5` and `IMO_1959_p1` in the Appendix.

## 5 Conclusion

In this paper, we introduced Lyra, a novel pipeline that takes advantage of *Tool Correction* and *Conjecture Correction*. *Tool Correction* employs prior knowledge to employ predefined tools (e.g. sledgehammer) to guide incorrect tool replacement. *Conjecture Correction*, interacting with the prover environment, integrates previous formal sketch and prover error messages for better sketch generation. We demonstrated the feasibility and effectiveness of Lyra by reaching state-of-the-art performance 55.3% and 51.2% on the miniF2F dataset validation and test, respectively, with the Isabelle theorem prover. Central to our method is the incorporation of prior knowledge and the development of a comprehensive GPT-4 refinement framework. Our ablations showed that both *Tool Correction* and *Conjecture Correction* are critical to the success of Lyra.

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

# APPENDIX

## A  BASELINE

To evaluate the effectiveness of our proposed methodology, we employed several baseline methods, which follow previous work setting (Jiang et al., 2023; Zhao et al., 2023).

**Sledgehammer with heuristics** The first baseline is Sledgehammer (Paulson, 2010), which a proof automation tool in the Isabelle environment. Additionally, we utilized Sledgehammer supplemented with heuristics, which integrates 11 prevalent tactics (i.e., auto, simp, blast, fastforce, force, evapl, presburger, sos, arith, linarith, auto simp: field simps) with Sledgehammer. If all tactics fail, the system employs Sledgehammer (Jiang et al., 2023).

**Language model based methods** Thor (Jiang et al., 2022) combines language models with automatic theorem provers to hep select premises from a vast library. Thor+expert (Wu et al., 2022) iteration enhances a neural theorem prover by training it on theorems that have been successfully formalized. Draft, Sketch, and Prove (DSP) (Jiang et al., 2023) transforms informal proofs into formal sketches and utilizes these formal sketches to guide an automated prover. Another LLM-based method is Subgoal-Proof Learning (Zhao et al., 2023), which incorporates subgoal proof to replace informal proof and proposes a prompt selection framework.

Following previous work (Jiang et al., 2023; Zhao et al., 2023), we excluded representative methods such as HyperTree Proof Search (HTPS) (Lample et al., 2022) and GPT-f with expert iteration (Polu et al., 2023), which are implemented using Lean de Moura et al. (2015), a different interactive theorem prover. The disparity in tactics and automation between Lean and Isabelle renders them not directly comparable to our method.

## B  NUMBER OF FIXED WRONG STEPS BY TOOL CORRECTION

To prove the effectiveness of Tool Correction, we calculate how many fixed wrong steps by Tool Correction, compared to baseline sledgehammer + heuristics.

Table 3: **The number of wrong fixed steps by Tool Correction**.

| Dataset | Sledehammer+heuristics | Tool Correction | Number of Fixed Wrong Steps |
|---------|------------------------|-----------------|-----------------------------|
| miniF2F-valid | 2260 | 3486 | 1226 |
| miniF2F-test | 2594 | 3887 | 1293 |

Proof comes from: the miniF2F validation set (pass rate 55.3%) and test set result (pass rate 51.2%), with GPT-4, human informal proof, Conjecture Correction and Tool Correction. The definition of proof step: a proof step is regarded as a tactic. Calculation protocol: if sledgehammer+heuristic or Tool Correction fails to validate the current tactic, then the current proving process will be terminated and we will turn to the next formal proof validation. Finally, we calculate how many correct tactics/proof steps. On miniF2F-valid, sledgehammer+heuristics can help the prover successfully pass 2260 steps. After adding Tool Correction, the number increases to 3486 steps. Therefore, Tool Correction fixes 1226 wrong steps. On miniF2F-test, Tool Correction fixes 1293 wrong steps.

## C  RELATION BETWEEN PERFORMANCE AND THE TIME LIMIT

Given a time limit, the performance of Lyra is related to whether allowing parallel process. If allows the parallel process, then we can keep the performance (55.3% on validation and 51.2% on test), if the time limit is larger or equal to 10 minutes. For the relationship between time limit and performance, we can refer to Figure 2 on Page 8, which presents the relationship between the number of attempts and the performance. One attempt takes 2 minutes, if not allow the parallel process.

Usually, it takes about 1 to 2 minutes to finish one attempt (we take 2 min/attempt here), where each problem is allowed to try 200 attempts(a total of 400 minutes), in our setting. For DSP, the 200 attempts can be processed in parallel. Hence, if allowing the parallel process, the maximum time cost

Table 4: **The relation between performance and time limit**.

| Parallel Process | Time Limit | Method | Time Cost | miniF2F-valid | miniF2F-test |
|---|---|---|---|---|---|
| ✗ | 10 mins | DSP | 10 mins | 32.7% | 25.4% |
| ✗ | 10 mins | Lyra | 10 mins | 33.6% | 28.6% |
| ✗ | 30 mins | DSP | 30 mins | 40.1% | 31.5% |
| ✗ | 30 mins | Lyra | 30 mins | 40.1% | 36.0% |
| ✗ | 400 mins | DSP | 400 mins | 50.4% | 42.6% |
| ✗ | 400 mins | Lyra | 400 mins | 55.3% | 51.2% |
| ✗ | 600 mins | DSP | 400 mins | 50.4% | 42.6% |
| ✗ | 600 mins | Lyra | 400 mins | 55.3% | 51.2% |
| ✓ | 10 mins | DSP | 2 mins | 50.4% | 42.6% |
| ✓ | 10 mins | Lyra | 10 mins | 55.3% | 51.2% |
| ✓ | 10 mins | DSP | 2 mins | 50.4% | 42.6% |
| ✓ | 10 mins | Lyra | 10 mins | 55.3% | 51.2% |

is 2 minutes. For Lyra, these 200 attempts are divided into 40 patches, where each patch contains five attempts. The 40 patches can be processed in parallel. Hence, if allowing the parallel process, the maximum time cost is 10 minutes. If not allowing the parallel process, when the time limit is 10 mins, Lyra achieves 33.6% miniF2F-validation and 28.6% miniF2F-test. And if not allow the parallel process, when the time limit is 30 mins, Lyra achieves 40.1% miniF2F-validation and 36.0% miniF2F-test.

## D    WHY THE INFORMAL PROOF AND THE FORMAL SKETCH ARE ACTUALLY QUITE DIFFERENT

The informal proof is a guide to formal proof, and it is not necessary that the informal proof steps and formal proof steps are one-to-one. The following two QAs may help better understand the explanation.

Why Informal proof and the formal sketch are quite different? Reason: The informal proof is a guide to formal proof generation. According to the original DSP paper, the informal proof only needs to be useful for producing a sketch in the next stage.

Why the formal one does not cover continuity or limit? Reason: Actually, the formal one covers the limit. For example, the formal proof has shown that "ultimately have "1 < ?S" by simp [ATPWithTC]", which covers the limit. The limit is proved via the ATP and Tool Correction.

## E    HYPOTHESES FOR WHY THE LLMS ARE EVEN ABLE TO UNDERSTAND ISABELLE ERROR MESSAGES

There are two hypotheses for why the LLMs are even able to understand Isabelle error messages.

Let's start from an easy point: the LLMs can understand English and Isabelle syntax. LLMs can understand English, which is approved by OpenAI. Therefore, this conclusion is obvious. Also, LLMs can understand Isabelle syntex. There are publicly available Isabelle proof corpus. For example, The entire AFP library, the largest formal library that contains most of Isabelle proofs, is 180MB in size (Wu et al., 2022). LLMs can understand Isabelle's syntax, and this is proved by previous works (Wu et al., 2022; Jiang et al., 2022; 2023; Zhao et al., 2023).

**Hypothesis 1: The publicly available Isabelle proof corpus also has examples of error messages**. The training dataset of GPT-3 (Brown et al., 2020) contains Common Crawl datasets (Raffel et al., 2020), which contain Stackflow. On the Stackflow, we find examples of error messages. Example 1:

Failed to apply initial proof method: using this: [ ] ∈ ns_public goal (1 subgoal): 1. ∀ A B X. Says A B X ∉ set_of_list [ ]. Example 2: Failed to apply proof method: using this: (y, x) ∈ r$\hat{*}$ (z, y) ∈ r goal (1 subgoal): 1. (z, x) ∈ r$\hat{*}$. Example 3: Failed to apply initial proof method: using this: n <a n <b goal (1 subgoal): 1. n * n <a * b. Therefore, the publicly available Isabelle proof corpus may also have examples of error messages.

**Hypothesis 2: the LLMs may understand Isabelle's error message if they understand English and Isabelle's syntax, but do not have to see Isabelle's error messages before**. First, we show what Isabelle's error messages look like. The error message is written in English, such as "Failed to apply proof method using this: 0 <y goal (1 subgoal): 1. 9 * (x * sin x) + 4 / (x * sin x) = (9 * (x * sin x)<ŝup>2 + 4) / (x * sin x) At command "by" ". According to the error message, we can find that the error message only contains English words and Isabelle syntax (such as <ŝup>). Therefore, to understand Isabelle's error message, the proposed LLM may not need to see Isabelle's error message before, but just has to understand English and Isabelle's syntax.

## F  TOOL CORRECTION GENERALIZATION

We discuss the Tool Correction generalization in this section.

### F.1  ADDITIONAL EXPERIMENT

To prove the generalization of Tool Correction, we combine DT-Solver (Wang et al., 2023b) with Tool Correction on PISA dataset (Jiang et al., 2021).

- Why not apply Lyra on the PISA dataset? Similarly to the experiment setting of DSP and Sub-goal learning, the Lyra needs informal information, such as informal problem and informal proof. To the best of our knowledge, only miniF2F has a Isabelle dataset that has informal information, while PISA does not.

- Why choose DT-Solver? We used to reproduce the results of DT-Solver so that the experiment can be finished before the rebuttal deadline. messages to refine proof.

- Why We choose the PISA dataset? It is used to evaluate previous famous works, such as Lisa (Jiang et al., 2021), Thor (Jiang et al., 2022) and Thor + expert iteration (Wu et al., 2022).

Table 5: **The experiment result of DT-Solver with Tool Correction on PISA dataset**.

| Dataset | DT-Solver | DT-Solver + sledgehammer+heuristics | DT-Solver + Tool Correction |
|---------|-----------|-------------------------------------|-----------------------------|
| PISA    | 37.0%     | 37.0%                               | 55.2%                       |

According to the experiment results, we can observe that

- Tool Correction significantly improves the performance, from 37.0 to 55.2%. This proves that Tool Correction is also useful for other datasets, such as the PISA dataset.

- The sledgehammer+heuristics does not work well for DT-Solver on the PISA dataset, because DT-Solver hardly uses sledgehammer tactics, while sledgehammer+heuristics only works when the given tactics is sledgehammer.

- We believe the above experiment can prove that TC is also useful for other datasets but not limited to miniF2F, even for different models. Meanwhile, TC is a plug-in-plug-out module, which is easily integrated into other methods.

### F.2  TOOL CORRECTION TELLS US SOMETHING FUNDAMENTAL ABOUT THE NATURE OF MATHEMATICAL PROOFS

Tool Correction tells us something fundamental about the nature of mathematical proofs: try as many proving tools as we can, so that we can improve the performance by reducing the occurrence of "the conjecture is correct, but the proving tools fail to prove the conjecture".

Why does the simple strategy used by TC, i.e., iteratively replacing the tactics in a failed proof with one of the 11 tactics from the tool heuristics set, work?

- Recall when can we successfully prove a conjecture
    - The conjecture should be correct, and the proving tools (such as a sledgehammer) can prove the conjecture.

- Recall when we fail to prove a conjecture: the proving tools fail to prove the conjecture.
    - Situation 1: The conjecture is incorrect.

    - Situation 2: Or, the correction is correct, but the proving tool is not correct so the proving tool cannot prove the conjecture.

- Why Observation: prover (such as Isabelle or Lean) fails to validate a formal proof, which may caused by "the conjecture is correct, but the proving tools fail to prove the conjecture". The following is an example.
    - Conjecture: 1*1=1

    - If the proving tool can not process multiplication operation, then it will fail to prove the conjecture 1*1=1, though the conjecture is correct.

- Therefore, to improve the performance, we can reduce the occurrence of "the conjecture is correct, but the proving tools fail to prove the conjecture"
    - One solution is Tool Correction: try as many proving tools as we can to prove the conjecture.

### F.3 OTHER POTENTIAL TACTICS BESIDES THE SLEDGEHAMMER + 11 TACTICS

To improve the effectiveness of TC, we can easily add tactics, including `by clarify`, `by fact`, `by rule`, `by erule`, `by elim`, `by induction`, `by algebra` and so on.

According to the DSP paper, the 11 tactics can be used to process algebra and number theory problems. And the tactics also have their own advantages and disadvantages. For example, it seems that `by simp` is not very good at processing natural numbers, as shown in Figure 1 in our paper.

As introduced in the DSP paper, the Sledgehammer (Paulson, 2010) is relatively special, as it works by flattening the goals encoded in the higher-order logic used by Isabelle/HOL into other logics (e.g., first-order logic) which can then be fed into automated theorem provers. The development of proving tools is also an important direction of automated theorem proving. Recently, there has been a work named Magnushammer Mikuła et al. (2023), which is more powerful than Sledgehammer.

## G   IMO 1974 P5 CASE

**Statement:** Determine all possible values of $S = \frac{a}{a+b+d} + \frac{b}{a+b+c} + \frac{c}{b+c+d} + \frac{d}{a+c+d}$ where $a, b, c, d$, are arbitrary positive numbers.

**Informal Proof (Human):**
Note that $2 = \frac{a}{a+b} + \frac{b}{a+b} + \frac{c}{c+d} + \frac{d}{c+d} > S > \frac{a}{a+b+c+d} + \frac{b}{a+b+c+d} + \frac{c}{a+b+c+d} + \frac{d}{a+b+c+d} = 1$.
We will now prove that $S$ can reach any range in between 1 and 2.
Choose any positive number $a$. For some variables such that $k, m, l > 0$ and $k + m + l = 1$, let $b = ak$, $c = am$, and $d = al$. Plugging this back into the original fraction, we get $S = \frac{a}{a+ak+al} + \frac{ak}{a+ak+am} + \frac{am}{ak+am+al} + \frac{al}{a+am+al} = \frac{1}{1+k+l} + \frac{k}{1+k+m} + \frac{m}{k+m+l} + \frac{l}{1+m+l}$. The above equation can be further simplified to $S = \frac{1}{2-m} + \frac{k}{2-l} + m + \frac{l}{2-k}$. Note that $S$ is a continuous function and that $f(m) = m + \frac{1}{2-m}$ is a strictly increasing function. We can now decrease $k$ and $l$ to make $m$ tend arbitrarily close to 1. We see $\lim_{m \to 1} m + \frac{1}{2-m} = 2$, meaning $S$ can be brought arbitrarily close to 2. Now, set $a = d = x$ and $b = c = y$ for some positive real numbers $x, y$. Then $S = \frac{2x}{2x+y} + \frac{2y}{2y+x} = \frac{2y^2+8xy+2x^2}{2y^2+5xy+2x^2}$. Notice that if we treat the numerator and denominator each as a quadratic in $y$, we will get $1 + \frac{g(x)}{2y^2+5xy+2x^2}$, where $g(x)$ has a degree lower than 2. This means taking $\lim_{y \to \infty} 1 + \frac{g(x)}{2y^2+5xy+2x^2} = 1$, which means $S$ can be brought arbitrarily close to 1. Therefore, we are done.

**Formal Proof:**
```
theorem imo_1974_p5:
  fixes a b c d s :: real
  assumes "a>0" "b>0" "c>0" "d>0"
  assumes h0 : "s=a/(a+b+d) + b/(a+b+c) + c/(b+c+d) + d/(a+c+d)"
  shows "1<s \<and> s<2"
proof -
  let ?S = "a/(a+b+d) + b/(a+b+c) + c/(b+c+d) + d/(a+c+d)"
  have c0: "a/(a+b)+b/(a+b)+c/(c+d)+d/(c+d) = 2" using assms by auto [ATPWithTC]
  have c1: "a/(a+b+c+d)+b/(a+b+c+d)+c/(a+b+c+d)+d/(a+b+c+d) = 1" using assms by
      auto [ATPWithTC]
  have "1 < ?S" using assms by (simp add: add_pos_pos c1) [ATPWithTC]
  moreover have "?S < 2" using assms by (simp add: add_pos_pos c0) [ATPWithTC]
  then show "1<s \<and> s<2" using assms h0 by auto [ATPWithTC]
qed
```

Figure 4: `IMO_1974_p5:  first round.`

**Statement:** Determine all possible values of $S = \frac{a}{a+b+d} + \frac{b}{a+b+c} + \frac{c}{b+c+d} + \frac{d}{a+c+d}$ where $a, b, c, d$, are arbitrary positive numbers.

**Informal Proof (Human):**

Note that $2 = \frac{a}{a+b} + \frac{b}{a+b} + \frac{c}{c+d} + \frac{d}{c+d} > S > \frac{a}{a+b+c+d} + \frac{b}{a+b+c+d} + \frac{c}{a+b+c+d} + \frac{d}{a+b+c+d} = 1$.
We will now prove that $S$ can reach any range in between 1 and 2.
Choose any positive number $a$. For some variables such that $k, m, l > 0$ and $k + m + l = 1$, let $b = ak$, $c = am$, and $d = al$. Plugging this back into the original fraction, we get $S = \frac{a}{a+ak+al} + \frac{ak}{a+ak+am} + \frac{am}{ak+am+al} + \frac{al}{a+am+al} = \frac{1}{1+k+l} + \frac{k}{1+k+m} + \frac{m}{k+m+l} + \frac{l}{1+m+l}$. The above equation can be further simplified to $S = \frac{1}{2-m} + \frac{k}{2-l} + m + \frac{l}{2-k}$. Note that $S$ is a continuous function and that $f(m) = m + \frac{1}{2-m}$ is a strictly increasing function. We can now decrease $k$ and $l$ to make $m$ tend arbitrarily close to 1. We see $\lim_{m \to 1} m + \frac{1}{2-m} = 2$, meaning $S$ can be brought arbitrarily close to 2. Now, set $a = d = x$ and $b = c = y$ for some positive real numbers $x, y$. Then $S = \frac{2x}{2x+y} + \frac{2y}{2y+x} = \frac{2y^2 + 8xy + 2x^2}{2y^2 + 5xy + 2x^2}$. Notice that if we treat the numerator and denominator each as a quadratic in $y$, we will get $1 + \frac{g(x)}{2y^2 + 5xy + 2x^2}$, where $g(x)$ has a degree lower than 2. This means taking $\lim_{y \to \infty} 1 + \frac{g(x)}{2y^2 + 5xy + 2x^2} = 1$, which means $S$ can be brought arbitrarily close to 1. Therefore, we are done.

**Formal Proof:**

```
theorem imo_1974_p5:
  fixes a b c d s :: real
  assumes "a>0" "b>0" "c>0" "d>0"
  assumes h0 : "s=a/(a+b+d) + b/(a+b+c) + c/(b+c+d) + d/(a+c+d)"
  shows "1<s \<and> s<2"
proof -
  let ?S = "a/(a+b+d) + b/(a+b+c) + c/(b+c+d) + d/(a+c+d)"

  (* Let's show that S > 1 *)
  have c1: "a/(a+b+c+d) + b/(a+b+c+d) + c/(a+b+c+d) + d/(a+b+c+d) = 1"
    using assms by (simp add: divide_simps)
  have "1 < ?S"
  proof -
    note add_strict_mono[of a a b d, of b a c b, of c b d c, of d a c d]
    then have "a/(a+b+d) > a/(a+b+c+d)" "b/(a+b+c) > b/(a+b+c+d)"
             "c/(b+c+d) > c/(a+b+c+d)" "d/(a+c+d) > d/(a+b+c+d)"
      using assms by (simp_all add: divide_strict_right_mono) [ATPWithTC]
    then show "1 < ?S" using c1 by auto [ATPWithTC]
  qed

  (* Let's show that S < 2 *)
  have c2: "a/(a+b) + b/(a+b) + c/(c+d) + d/(c+d) = 2"
    using assms by (simp add: divide_simps) [ATPWithTC]
  have "?S < 2"
  proof -
    note add_strict_mono[of a a b 0, of b a 0 c, of c 0 d b, of d a 0 c]
    then have "a/(a+b+d) < a/(a+b)" "b/(a+b+c) < b/(a+b)"
             "c/(b+c+d) < c/(c+d)" "d/(a+c+d) < d/(c+d)"
      using assms by (simp_all add: divide_strict_right_mono) [ATPWithTC]
    then show "?S < 2" using c2 by auto [ATPWithTC]
  qed

  then show "1<s \<and> s<2" using assms h0 `1 < ?S` by auto [ATPWithTC]
qed
```

Figure 5: `IMO_1974_p5:  second round.`

**Statement:** Determine all possible values of $S = \frac{a}{a+b+d} + \frac{b}{a+b+c} + \frac{c}{b+c+d} + \frac{d}{a+c+d}$ where $a, b, c, d,$ are arbitrary positive numbers.

**Informal Proof (Human):**

Note that $2 = \frac{a}{a+b} + \frac{b}{a+b} + \frac{c}{c+d} + \frac{d}{c+d} > S > \frac{a}{a+b+c+d} + \frac{b}{a+b+c+d} + \frac{c}{a+b+c+d} + \frac{d}{a+b+c+d} = 1$. We will now prove that $S$ can reach any range in between 1 and 2. Choose any positive number $a$. For some variables such that $k, m, l > 0$ and $k + m + l = 1$, let $b = ak$, $c = am$, and $d = al$. Plugging this back into the original fraction, we get $S = \frac{a}{a+ak+al} + \frac{ak}{a+ak+am} + \frac{am}{ak+am+al} + \frac{al}{a+am+al} = \frac{1}{1+k+l} + \frac{k}{1+k+m} + \frac{m}{k+m+l} + \frac{l}{1+m+l}$. The above equation can be further simplified to $S = \frac{1}{2-m} + \frac{k}{2-l} + m + \frac{1}{2-k}$. Note that $S$ is a continuous function and that $f(m) = m + \frac{1}{2-m}$ is a strictly increasing function. We can now decrease $k$ and $l$ to make $m$ tend arbitrarily close to 1. We see $\lim_{m \to 1} m + \frac{1}{2-m} = 2$, meaning $S$ can be brought arbitrarily close to 2. Now, set $a = d = x$ and $b = c = y$ for some positive real numbers $x, y$. Then $S = \frac{2x}{2x+y} + \frac{2y}{2y+x} = \frac{2y^2+8xy+2x^2}{2y^2+5xy+2x^2}$. Notice that if we treat the numerator and denominator each as a quadratic in $y$, we will get $1 + \frac{g(x)}{2y^2+5xy+2x^2}$, where $g(x)$ has a degree lower than 2. This means taking $\lim_{y \to \infty} 1 + \frac{g(x)}{2y^2+5xy+2x^2} = 1$, which means $S$ can be brought arbitrarily close to 1. Therefore, we are done.

**Formal Proof:**

```
(*statement begin*)
theorem imo_1974_p5:
  fixes a b c d s :: real
  assumes "a>0" "b>0" "c>0" "d>0"
  assumes h0 : "s=a/(a+b+d) + b/(a+b+c) + c/(b+c+d) + d/(a+c+d)"
  shows "1<s \<and> s<2"
(*statement end*)
proof -
(* Let's correct the proof. *)
  (* Define the sum S *)
  let ?S = "a/(a+b+d) + b/(a+b+c) + c/(b+c+d) + d/(a+c+d)"
  (* Show that S > 1 *)
  have "a/(a+b+c+d) + b/(a+b+c+d) + c/(a+b+c+d) + d/(a+b+c+d) = 1"
    using assms by (simp add: divide_simps) [ATPWithTC]
  moreover have "?S > a/(a+b+c+d) + b/(a+b+c+d) + c/(a+b+c+d) + d/(a+b+c+d)"
  proof -
    have "a/(a+b+d) > a/(a+b+c+d)" "b/(a+b+c) > b/(a+b+c+d)"
          "c/(b+c+d) > c/(a+b+c+d)" "d/(a+c+d) > d/(a+b+c+d)"
      using assms by (simp_all add: divide_strict_right_mono) [ATPWithTC]
    then show ?thesis by auto [ATPWithTC]
  qed
  ultimately have "1 < ?S" by simp [ATPWithTC]
  (* Show that S < 2 *)
  have "a/(a+b) + b/(a+b) + c/(c+d) + d/(c+d) = 2"
    using assms by (simp add: divide_simps) [ATPWithTC]
  moreover have "?S < a/(a+b) + b/(a+b) + c/(c+d) + d/(c+d)"
  proof -
    have "a/(a+b+d) < a/(a+b)" "b/(a+b+c) < b/(a+b)"
          "c/(b+c+d) < c/(c+d)" "d/(a+c+d) < d/(c+d)"
      using assms by (simp_all add: divide_strict_right_mono) [ATPWithTC]
    then show ?thesis by auto [ATPWithTC]
  qed
  ultimately have "?S < 2" by simp [ATPWithTC]
  (* Conclude the proof *)
  then show "1<s \<and> s<2" using assms h0 `1 < ?S` by auto [ATPWithTC]
qed
end
```

Figure 6: `IMO_1974_p5`:  Third round.

# H    IMO 1959 P1 CASE

**Statement:** Prove that the fraction $\frac{21n+4}{14n+3}$ is irreducible for every natural number $n$.

- - - - - - - - - - - - - - - - - - - - - - - - - - - - - - - - - - - - - - - - - -

**Informal Proof (GPT-4):** We can use the Euclidean algorithm to prove this. If the greatest common divisor (gcd) of the numerator and the denominator is 1, then the fraction is irreducible.

Let's find the gcd of $21n + 4$ and $14n + 3$.

We start by subtracting the smaller number from the larger one. We get: $(21n+4)-(14n+3) = 7n+1$.

Next, we subtract $7n + 1$ from $14n + 3$. We get: $(14n + 3) - (7n + 1) = 7n + 2$.

Now, we subtract $7n + 1$ from $7n + 2$. We get: $(7n + 2) - (7n + 1) = 1$.

Since the greatest common divisor of $21n + 4$ and $14n + 3$ is 1, the fraction $\frac{21n+4}{14n+3}$ is irreducible for every natural number $n$.

- - - - - - - - - - - - - - - - - - - - - - - - - - - - - - - - - - - - - - - - - -

**Formal Proof:**

```
theorem imo_1959_p1:
  fixes n :: nat
  shows "gcd (21*n + 4) (14*n + 3) = 1"
proof -
  (* Let's find the gcd of 21n + 4 and 14n + 3. *)
  have h1: "gcd (21*n + 4) (14*n + 3) = gcd ((21*n + 4) - (14*n + 3)) (14*n + 3)"
    by (metis add_diff_inverse_nat gcd_diff1_nat le_add2 nat_le_linear) [
        ATPWithTC]
  (* We start by subtracting the smaller number from the larger one.
  We get: (21n + 4) - (14n + 3) = 7n + 1. *)
  also have "... = gcd (7*n + 1) (14*n + 3)" by simp [ATPWithTC]
  (* Next, we subtract 7n + 1 from 14n + 3. We get: (14n + 3) - (7n + 1) = 7n + 2.
     *)
  also have "... = gcd (7*n + 1) (7*n + 2)"
    by (metis add_diff_inverse_nat gcd_diff1_nat le_add2 nat_le_linear) [
        ATPWithTC]
  (* Now, we subtract 7n + 1 from 7n + 2. We get: (7n + 2) - (7n + 1) = 1. *)
  also have "... = gcd (7*n + 1) 1" by simp [ATPWithTC]
  (* Since 1 is the greatest common divisor of any number,
  we find that gcd (7*n + 1) 1 = 1. *)
  finally show ?thesis by simp [ATPWithTC]
qed
```

Figure 7: `IMO_1959_p1: first round`.

**Statement:** Prove that the fraction $\frac{21n+4}{14n+3}$ is irreducible for every natural number $n$.

- - - - - - - - - - - - - - - - - - - - - - - - - - - - - - - - - - - - - - - - - - - - - - -

**Informal Proof (GPT-4):**

We can use the Euclidean algorithm to prove this. If the greatest common divisor (gcd) of the numerator and the denominator is 1, then the fraction is irreducible.

Let's find the gcd of $21n + 4$ and $14n + 3$.

We start by subtracting the smaller number from the larger one. We get: $(21n+4)-(14n+3) = 7n+1$.

Next, we subtract $7n + 1$ from $14n + 3$. We get: $(14n + 3) - (7n + 1) = 7n + 2$.

Now, we subtract $7n + 1$ from $7n + 2$. We get: $(7n + 2) - (7n + 1) = 1$.

Since the greatest common divisor of $21n + 4$ and $14n + 3$ is 1, the fraction $\frac{21n+4}{14n+3}$ is irreducible for every natural number $n$.

- - - - - - - - - - - - - - - - - - - - - - - - - - - - - - - - - - - - - - - - - - - - - - -

**Formal Proof:**

```
theorem imo_1959_p1:
  fixes n :: nat
  shows "gcd (21*n + 4) (14*n + 3) = 1"
proof -
(* Let's find the gcd of 21n + 4 and 14n + 3. *)
  have h1: "gcd (21*n + 4) (14*n + 3) = gcd (14*n + 3) ((21*n + 4) mod (14*n + 3)
      )"
    using gcd_red_nat by blast [ATPWithTC]
  (* We compute modulo operation and get: (21n + 4)mod(14n + 3) = 7n + 1 *)
  also have "... = gcd (14*n + 3) (7*n + 1)"
    by (smt add.right_neutral add_Suc_right add_lessD1 gcd.commute mult_2
        mult_Suc
    mult_less_cancel1 mult_numeral_1 numeral_Bit0 numeral_Bit1 numeral_eq_iff
    semiring_norm(76) semiring_norm(78) semiring_norm(8)) [ATPWithTC]
  (* Then we again compute the GCD with the modulo operation. *)
  also have "... = gcd (7*n + 1) ((14*n + 3) mod (7*n + 1))"
    using gcd_red_nat by blast [ATPWithTC]
  (* We compute the modulo operation and get: (14n + 3)mod(7n + 1) = n + 2. *)
  also have "... = gcd (7*n + 1) (n + 2)"
    by (smt add_2_eq_Suc add_lessD1 gcd.commute mult_2 mult_Suc mult_less_cancel1
    semiring_norm(76) semiring_norm(78) semiring_norm(8)) [ATPWithTC]
  (* Then we again compute the GCD with the modulo operation. *)
  also have "... = gcd (n + 2) ((7*n + 1) mod (n + 2))"
    using gcd_red_nat by blast [ATPWithTC]
  (* We compute the modulo operation and get: (7n + 1)mod(n + 2) = 1. *)
  also have "... = gcd (n + 2) 1"
    by (smt add.commute add_lessD1 gcd.commute mult_less_cancel1
    semiring_norm(76) semiring_norm(78) semiring_norm(8)) [ATPWithTC]
  (* gcd of any number with 1 is 1. *)
  also have "... = 1" by simp [ATPWithTC]
  finally show ?thesis . [ATPWithTC]
qed
```

Figure 8: `IMO_1959_p1:  second round.`

**Statement:** Prove that the fraction $\frac{21n+4}{14n+3}$ is irreducible for every natural number $n$.

---

**Informal Proof (GPT-4):**

We can use the Euclidean algorithm to prove this. If the greatest common divisor (gcd) of the numerator and the denominator is 1, then the fraction is irreducible.

Let's find the gcd of $21n + 4$ and $14n + 3$.

We start by subtracting the smaller number from the larger one. We get: $(21n+4)-(14n+3) = 7n+1$.

Next, we subtract $7n + 1$ from $14n + 3$. We get: $(14n + 3) - (7n + 1) = 7n + 2$.

Now, we subtract $7n + 1$ from $7n + 2$. We get: $(7n + 2) - (7n + 1) = 1$.

Since the greatest common divisor of $21n + 4$ and $14n + 3$ is 1, the fraction $\frac{21n+4}{14n+3}$ is irreducible for every natural number $n$.

---

**Formal Proof:**

```
theorem imo_1959_p1:
  fixes n :: nat
  shows "gcd (21*n + 4) (14*n + 3) = 1"
theorem imo_1959_p1:
  fixes n :: nat
  shows "gcd (21*n + 4) (14*n + 3) = 1"
proof -
(* Let's find the gcd of 21n + 4 and 14n + 3. *)
  have h1: "gcd (21*n + 4) (14*n + 3) = gcd (14*n + 3) ((21*n + 4) mod (14*n + 3)
      )"
    using gcd_red_nat by blast [ATPWithTC]
  (* We compute modulo operation and get: (21n + 4)mod(14n + 3) = 7n + 1 *)
  also have "... = gcd (14*n + 3) (7*n + 1)"
    by (simp add: mod_simps) [ATPWithTC]
  (* Then we again compute the GCD with the modulo operation. *)
  also have "... = gcd (7*n + 1) ((14*n + 3) mod (7*n + 1))"
    using gcd_red_nat by blast [ATPWithTC]
  (* We compute the modulo operation and get: (14n + 3)mod(7n + 1) = n + 2. *)
  also have "... = gcd (7*n + 1) (n + 2)"
    by (simp add: mod_simps) [ATPWithTC]
  (* Then we again compute the GCD with the modulo operation. *)
  also have "... = gcd (n + 2) ((7*n + 1) mod (n + 2))"
    using gcd_red_nat by blast [ATPWithTC]
  (* We compute the modulo operation and get: (7n + 1)mod(n + 2) = 1. *)
  also have "... = gcd (n + 2) 1"
    by (simp add: mod_simps) [ATPWithTC]
  (* gcd of any number with 1 is 1. *)
  also have "... = 1" by simp [ATPWithTC]
  finally show ?thesis . [ATPWithTC]
qed
```

Figure 9: `IMO_1959_p1:` third round.

**Statement:** Prove that the fraction $\frac{21n+4}{14n+3}$ is irreducible for every natural number $n$.

**Informal Proof (GPT-4):**
We can use the Euclidean algorithm to prove this. If the greatest common divisor (gcd) of the numerator and the denominator is 1, then the fraction is irreducible.
Let's find the gcd of $21n + 4$ and $14n + 3$.
We start by subtracting the smaller number from the larger one. We get: $(21n+4)-(14n+3) = 7n+1$.
Next, we subtract $7n + 1$ from $14n + 3$. We get: $(14n + 3) - (7n + 1) = 7n + 2$.
Now, we subtract $7n + 1$ from $7n + 2$. We get: $(7n + 2) - (7n + 1) = 1$.
Since the greatest common divisor of $21n + 4$ and $14n + 3$ is 1, the fraction $\frac{21n+4}{14n+3}$ is irreducible for every natural number $n$.

**Formal Proof:**

```
theorem imo_1959_p1:
  fixes n :: nat
  shows "gcd (21*n + 4) (14*n + 3) = 1"
proof -
(* We can use the Euclidean algorithm to prove this. *)
  (* If the greatest common divisor (gcd) of the numerator and the denominator is
       1,
  then the fraction is irreducible. *)
  have h1: "gcd (21*n + 4) (14*n + 3) = gcd (14*n + 3) ((21*n + 4) mod (14*n + 3)
      )"
    using gcd_red_nat by blast [ATPWithTC]
  (* We start by subtracting the smaller number from the larger one. *)
  also have "... = gcd (14*n + 3) ((21 - 14)*n + (4 - 3))"
    using mod_if by auto [ATPWithTC]
  (* We get: (21n + 4) − (14n + 3) = 7n + 1. *)
  also have "... = gcd (14*n + 3) (7*n + 1)"
    by simp [ATPWithTC]
  (* Next, we subtract 7n + 1 from 14n + 3. *)
  also have "... = gcd (7*n + 1) ((14*n + 3) mod (7*n + 1))"
    using gcd_red_nat by blast [ATPWithTC]
  (* We get: (14n + 3) − (7n + 1) = 7n + 2. *)
  also have "... = gcd (7*n + 1) ((14 - 7)*n + (3 - 1))"
    using mod_if by auto [ATPWithTC]
  also have "... = gcd (7*n + 1) (7*n + 2)"
    by simp [ATPWithTC]
  (* Now, we subtract 7n + 1 from 7n + 2. *)
  also have "... = gcd (7*n + 2) ((7*n + 1) mod (7*n + 2))"
    using gcd_red_nat by blast [ATPWithTC]
  (* We get: (7n + 2) − (7n + 1) = 1. *)
  also have "... = gcd (7*n + 2) (1)"
    using mod_if by auto [ATPWithTC]
  (* Since the greatest common divisor of 21n + 4 and 14n + 3 is 1, *)
  also have "... = 1"
    by simp [ATPWithTC]
  finally show ?thesis . [ATPWithTC]
qed
```

Figure 10: `IMO_1959_p1:  fourth round.`

## I IMO 1981 P6 CASE

**Statement:** The function $f(x, y)$ satisfies
(1) $f(0, y) = y + 1$,
(2) $f(x + 1, 0) = f(x, 1)$,
(3) $f(x + 1, y + 1) = f(x, f(x + 1, y))$,
for all non-negative integers $x, y$. Determine $f(4, 1981)$

- - - - - - - - - - - - - - - - - - - - - - - - - - - - - - - - - - - - - - - - - - - - - - - -

**Informal Proof (Human):**
We observe that $f(1, 0) = f(0, 1) = 2$ and that $f(1, y + 1) = f(1, f(1, y)) = f(1, y) + 1$, so by induction, $f(1, y) = y + 2$. Similarly, $f(2, 0) = f(1, 1) = 3$ and $f(2, y + 1) = f(2, y) + 2$, yielding $f(2, y) = 2y + 3$.
We continue with $f(3, 0) + 3 = 8$; $f(3, y + 1) + 3 = 2(f(3, y) + 3)$; $f(3, y) + 3 = 2^{y+3}$; and $f(4, 0) + 3 = 2^{2^2}$; $f(4, y) + 3 = 2^{f(4,y)+3}$.
It follows that $f(4, 1981) = 2^{2^{\cdot^{\cdot^{\cdot^2}}}} - 3$ when there are 1984 2s, Q.E.D.

- - - - - - - - - - - - - - - - - - - - - - - - - - - - - - - - - - - - - - - - - - - - - - - -

**Formal Proof:**

```
theorem imo_1981_p6:
  fixes f :: "nat \<Rightarrow> nat \<Rightarrow> nat"
  assumes "\<forall> y. f 0 y = y + 1"
    and "\<forall> x. f (x + 1) 0 = f x 1"
    and "\<forall> x y. f (x + 1) (y + 1) = f x (f (x + 1) y)"
  shows "\<forall> y. f 4 (y + 1) = 2^(f 4 y + 3) - 3"
(*statement end*)
proof -
  (* We observe that f(1,0) = f(0,1) = 2 *)
  have "\<forall>y. f 1 y = y + 2"
  proof
    fix y
    show "f 1 y = y + 2"
    proof (induction y)
      case 0
      then show ?case using assms(2)[rule_format, of "0"]
        by simp ATPWithTC]
    next
      case (Suc y)
      then show ?case using assms(3)[rule_format, of "0" "y"]
        by simp ATPWithTC]
    qed
  qed
  then have f1: "\<forall>y. f 1 y = y + 2"
    by simp
  (* Similarly, f(2,0) = f(1,1) = 3 and f(2,y+1) = f(2,y) + 2, yielding
      f(2,y) = 2y + 3. *)
  have "\<forall>y. f 2 y = 2*y + 3"
  proof
    fix y
    show "f 2 y = 2*y + 3"
    proof (induction y)
      case 0
      then show ?case using assms(2)[rule_format, of "1"]
        using f1 by simp ATPWithTC]
    next
      case (Suc y)
      then show ?case using assms(3)[rule_format, of "1" "y"]
        using f1 by simp ATPWithTC]
    qed
  qed
```

Figure 11: IMO_1981_p6: Part 1.

```
    then have f2: "\<forall>y. f 2 y = 2*y + 3"
      by simp [ATPWithTC]
    (* We continue with f(3, 0) + 3 = 8; f(3, y + 1) + 3 = 2(f(3, y) + 3);
         f(3, y) + 3 = 2^(y+3) *)
    have "\<forall>y. f 3 y + 3 = 2^(y + 3)"
    proof
      fix y
      show "f 3 y + 3 = 2^(y + 3)"
      proof (induction y)
        case 0
        then show ?case using assms(2)[rule_format, of "2"]
          using f2 by simp [ATPWithTC]
      next
        case (Suc y)
        then show ?case using assms(3)[rule_format, of "2" "y"]
          using f2 by simp [ATPWithTC]
      qed
    qed
    then have f3: "\<forall>y. f 3 y + 3 = 2^(y + 3)"
      by simp [ATPWithTC]
    (* and f(4, 0) + 3 = 2^{2^2}; f(4, y) + 3 = 2^{f(4,y)+3}. *)
    have "\<forall>y. f 4 (y + 1) = 2^(f 4 y + 3) - 3"
    proof
      fix y
      show "f 4 (y + 1) = 2^(f 4 y + 3) - 3"
      proof (induction y)
        case 0
        then show ?case using assms(2)[rule_format, of "3"]
          using f3 by simp [ATPWithTC]
      next
        case (Suc y)
        then show ?case using assms(3)[rule_format, of "3" "y"]
          using f3 by simp [ATPWithTC]
      qed
    qed
    then show ?thesis
      by simp [ATPWithTC]
  qed
```

Figure 12: `IMO_1981_p6: Part 2`. This problem is finished with human informal proof and *Tool Correction*, but without *Conjecture Correction*. Actually, if a problem can be solved by only *Tool Correction*, then it can be solved via *Tool Correction* and *Conjecture Correction*. The Lyra needs more attempts to solve the `IMO_1981_p6`, as *Conjecture Correction* needs more attempts.

