# OpenReview forum: "Lyra: Orchestrating Dual Correction in Automated Theorem Proving"
_ICLR.cc/2024/Conference — Submitted to ICLR 2024_

### Official Review · Reviewer_bwwN · 2023-10-21

**Soundness:** 3 good
**Presentation:** 3 good
**Contribution:** 3 good
**Rating:** 6
**Confidence:** 5

**Summary:**

This paper presents Lyra, an automated system integrating large language models for autoformalisation and symbolic tools such as sledgehammer in proof assistants. The two core components, Tool Correction and Conjecture Correction, are critical for its achieving SoTA performance on the miniF2F benchmark.

**Strengths:**

The proposed mechanism makes good intuitive sense. The performance improvement is impressive on the miniF2F benchmark. The ablation studies are well-presented and convincing.

**Weaknesses:**

There needs to be more contextualisation of the prior works. For the two major correction mechanisms, there have been direct prior works doing very similar or even identical things.
- Tool correction: in the DSP work [1], the authors already used sledgehammer + heuristics to close conjectures made. The understanding is that the Lyra method first tries a LLM-generated tactic to close conjectures, and if it doesn't work, try sledgehammer. This is largely similar and should be noted.
- In the Baldur work [2] from April 2023, the authors have proposed to use the proof assistant error message to repair the proofs. This is very similar to the conjecture correction with error messages and should be noted.

[1] Albert Qiaochu Jiang, Sean Welleck, Jin Peng Zhou, Timothee Lacroix, Jiacheng Liu, Wenda Li, Mateja Jamnik, Guillaume Lample, and Yuhuai Wu. Draft, sketch, and prove: Guiding formal theorem provers with informal proofs. In The Eleventh International Conference on Learning Representations, 2023.

[2] First, Emily, Markus N. Rabe, Talia Ringer, and Yuriy Brun. "Baldur: whole-proof generation and repair with large language models." arXiv preprint arXiv:2303.04910 (2023).

**Questions:**

None.

---

> ### Author Response · Authors · 2023-11-14
> **Response to Reviewer bwwN**
>
> Dear Reviewer bwwN,
>
> Thank you for appreciating our approach. We address your comments below.
>
> **Q1: Add the contribution of DSP.**
>
> A1: The part is shown on the first page of the Appendix. Because of the paper page limit, we have added the sledgehammer+heuristic description to the Appendix.
>
> Meanwhile, compared to sledgehammer+heuristics in DSP, our proposed Tool Correction has a different motivation and implementation. For example, if “by (simp add: div mult mod eq)” fails, the DSP will immediately stop validating, while our proposed Tool Correction will try all other potential tools. With Tool Correction, we would like to present one important point: the post-process of formal proof can significantly further improve performance, and future work could try to design a more powerful post-process method. And the following is the number of fixed wrong steps via Tool Correction.
>
> |  Dataset | Sledgehammer+heuristics   | Tool Correction  | Number of Fixed Steps|
> |  ---- |  ----  | ----  | ----  |
> | miniF2F-valid  |2260  | 3486 |1226 |
> | miniF2F-test  |2594  | 3887 |1293 |
>
> For more details, please kindly check our answer in  **The difference between Tool Correction and sledgehammer+heuristic** in **Author Response to All Reviewers**.
>
> **Q2: Baldur is very similar to the conjecture correction with error messages and should be noted.**
>
> A2: We have added Baldur to our reference. And the difference between Baldur and Lyra Conjection Correction is the following.
> Compared to Baldur which needs a training process, Lyra Conjection Correction is training-free, and it can reset the initial solution after several failure rounds. Please refer to the more detailed answer to **The difference between Lyra and other works** in the **Author response to all reviewers**.
>
>
> If you think our comment and update address your concerns, could you please consider raising your rating of the paper? Thank you!

---

> > ### Comment · Reviewer_bwwN · 2023-11-19
> >
> > Thank you for your reply. The explanation in the reply and the overall comment makes good sense.
> >
> > I shall retain my rating as I think it is what the contributions warrant.

---

> > > ### Author Response · Authors · 2023-11-21
> > > **Response to Reviewer bwwN**
> > >
> > > Dear Reviewer bwwN,
> > >
> > > Thank you very much for your reply. Thank you very much for your attention and support to this work, and wish you a good day.
> > >
> > > Best regards, Paper 1096 Authors

---

### Official Review · Reviewer_NmsF · 2023-10-31

**Soundness:** 3 good
**Presentation:** 3 good
**Contribution:** 2 fair
**Rating:** 6
**Confidence:** 3

**Summary:**

This paper presents a couple of approaches, namely Tool Correction (TC) and Conjecture Correction (CC), for improving the performance of LLM-guided autoformalization techniques based on the Draft-Sketch-Proof (DSP) paradigm. TC is a post-processing technique---once the LLM generates a formal proof sketch, the proof is fed to an Interactive Theorem Prover (ITP) and if the ITP returns an error, then TC uses simple heuristics to replace the tactics in the incorrect formal proof sketch one-by-one. CC, on the other hand, incorporates the error message from the ITP and generates a new prompt that includes the previous proof attempt and the error message. This interaction between the LLM and ITP is conductive up to 5 times. The proposed extensions to DSP are evaluated on the miniF2F benchmark and lead to improved proof success rates.

**Strengths:**

1. The techniques lead to an improvement over the state-of-the-art.
2. I find it a little surprising that models like GPT-4 and Codex are able to understand and act based on the error messages from Isabelle. Unfortunately, the paper does not explore this surprising observation in depth.

-----------------------------
Rating updated to 5 after discussion. Overall, I remain unconvinced by the technical contribution of TC but I find the empirical phenomenon of LLMs being able to interpret ITP error messages interesting

-----------------------------
Rating further updated after much discussion about the general applicability of TC to 6. In particular, the authors presented evidence that TC helps improve performance on another dataset, namely, LISA.

**Weaknesses:**

1. I am not an expert in the area so it is possible that the empirical results might be surprising to experts in a way that I am unable to appreciate. However, I find that the techniques used to achieve the state-of-the-art results do not involve any significant technical or empirical insights.

2. The heuristics used in TC seem too specific to the task and dataset. Are the heuristics used for TC transferable to other theorem proving tools and datasets? How do we know that these heuristics are not overfit to the miniF2f dataset? Do these heuristics represent some general insight into how mathematical theorems ought to be proved?

3. CC simply incorporates the ITP error message into the prompt. On a technical level, this is an obvious idea that has been tried before for proof generation [1] and code generation [2,3]. Some of these approaches require fine-tuning the model, so the scientific question to be evaluated here is does incorporating error feedback without any fine-tuning help **in general** for autoformalization. Since models such as Codex and GPT-4 are not explicitly trained on the error messages from ITPs, one would not expect simply providing the error messages in the prompts to be generally helpful. While the empirical results here suggest that error messages help when the ITP is Isabelle, it remains to be evaluated if it is helpful with other ITPs. It would also be useful to analyze the nature of the error messages generated by Isabelle. Do the improvements depend on the quality of the error messages? One would expect so but this would be another useful aspect to empirically evaluate.

[1] First, E., Rabe, M. N., Ringer, T., & Brun, Y. (2023). Baldur: whole-proof generation and repair with large language models. arXiv preprint arXiv:2303.04910.

[2] Le, H., Wang, Y., Gotmare, A. D., Savarese, S., & Hoi, S. C. H. (2022). Coderl: Mastering code generation through pretrained models and deep reinforcement learning. Advances in Neural Information Processing Systems, 35, 21314-21328.

[3] Wu, X., Cheriere, N., Zhang, C., & Narayanan, D. (2023). RustGen: An Augmentation Approach for Generating Compilable Rust Code with Large Language Models.

**Questions:**

1. Are the presented techniques overfit to miniF2F dataset and Isabelle? In particular, I am afraid this might be the case for TC.

2. Would CC work with the error messages from a different ITP? How much does the availability of formal proofs and error messages for a particular ITP affect the effectiveness of CC?  How much does the quality of the error message affect CC?

3. There are number of spelling errors in Algorithm 2.

---

> ### Author Response · Authors · 2023-11-14
> **Response to Reviewer NmsF (Part 1/2)**
>
> Dear Reviewer NmsF,
>
> Thank you for the detailed review. We will address your concerns below.
>
> **Q1: The technical or empirical insights.**
>
> A1: Tool Correction presents the importance of post-processing LLM response to improve performance, and Conjecture Correction proposes a framework that can extend the non-refinement framework to a refinement framework.
>
> The Insight of Tool Correction
> * Experiment Observation. LLM formal proof can not prove a simple conjecture x = 19∗(x div 19)+4. After our analysis, this is caused by LLM hallucination that LLM wrong believes by (simp add: div mult mod eq) can prove x = 19∗(x div 19)+4. To mitigate the hallucination, we propose to post-process the LLM-generated formal proof by predefined rules, which is called Tool Correction.
> * Following this direction,  the automated theorem proving performance can be further improved, if there are better rules to post-process the generated formal proof. And this is one of the important points that we want to present and prove by Tool Correction.
>
> The Insight of Conjecture Correction
> * The motivation of Conjecture Correction: propose a technique that can change a non-framework (such as DSP) to a refinement framework (such as Lyra), without too many modifications.
> * Experiment Observation. If directly appends the error message at the end of the prompt,  the LLM response is various. For example, after appending error at the end of (informal Proof 1, formal Proof 1, informal Proof 2, formal Promal 2) to get (informal Proof 1, formal Proof 1, informal Proof 2, formal Proof 2, error), and then ask LLM to refine formal Proof 2, the response may begin with LLM’s comments of formal proof 2, but not a formal proof.
> * Solution. Conjecture Correction adds “proof -” at the end of the instruction as an indicator. As most of the formal proofs begin with “proof -”, adding such an indicator could ask LLm to generate a formal proof.
>
> **Q2: Are the heuristics used for TC transferable to other theorem proving tools and datasets.**
>
> A2: The proposed Tool Correction can be transferable to other theorem proving tools and datasets.  The corresponding heuristics can be transferable to other datasets, and the heuristics need to be designed for different provers as they have different syntaxes.
>
> We discuss how can we transfer Tool Correction to other theorem proving tools, using Lean Prover as an example.
> * The Lean Prover has proving tools, such as simp, linarith, ring, and so on (these are its heuristics). When proving a conjecture, we can also try these different proving tools (heuristics) in Lean if the current tactic fails. Hence. Tool Correction is transferable to Lean Prover with Lean.
> * Tool Correction and heuristics are not related to datasets, but related to a prover (such as Isabelle or Lean). Hence, as long as we adapt Tool Correction to a prover, the proposed Tool Correction and heuristics can be transferable to other datasets.
> * Therefore, the Tool Correction and heuristics are transferable to other theorem proving tools datasets.
> * Finally, Tool Correction provides the following insights to overcome LLM hallucination. First, decompose the LLM response into different parts. Then, employs predefined rules to post-process the parts.
>
> **Q3: How do we know that these heuristics are not overfit to the miniF2f dataset?**
>
> A3: These heuristics are not overfit to the miniF2F dataset because they are related to a prover, but not a dataset.
>
> The proposed Tool Correction focuses on post-processing the formal proof which is written in Isabelle in this work. Therefore, the heuristics are related to Isabelle (in this work), but not any dataset, such as the miniF2F dataset.
>
> **Q4: Are the presented techniques overfit to miniF2F dataset and Isabelle? In particular, I am afraid this might be the case for TC.**
>
> A4: The presented techniques are not overfit to miniF2F or Isabelle.
>
> For Tool Correction,
> * We can easily adapt it to other provers, such as Lean. And this is illustrated in **Q2**.
> * Also, Tool Correction implementation is dataset-agnostic, and it is not related to any datasets, including miniF2F.
>
> For Conjecture Correction,
> * Conjecture Correction needs formal proof and the corresponding error message, which can come from Isabelle (in this work) and other provers (such as Lean)
> * Conjecture Correction is also not overfit to miniF2F, because its implementation is dataset-agnostic.

---

> > ### Author Response · Authors · 2023-11-14
> > **Response to Reviewer NmsF (Part 2/2)**
> >
> > **Q5: Do these heuristics represent some general insight into how mathematical theorems ought to be proved?**
> >
> > A5: Yes, these heuristics represent general insights into how mathematical theorems ought to be proved.
> >
> > As described in Question 1 (**Q1**),  Tool Correction suggests that: because of LLM hallucination, the prover validation may fail because employs incorrect tools, while the conjecture is correct. And we can partly solve the problem via post-process (such as replacing the incorrection tools), which is called Tool Correction in this work.  The automated theorem proving performance can be further improved, if there are better rules to post-process the generated formal proof. And this is one of the important points that we want to present and prove by Tool Correction.
> >
> >
> > **Q6: What is the difference between Lyra, Baldur, Coderl and RustGen?**
> >
> > A6: Compared to Baldur and Coderl, the Lyra is training-free. Compared with RustGen refining only with previous response and error message, the Lyra can sufficiently utilize problem information, informal proof information, previous response and error message. Meanwhile, Lyra will reset the initial solution, if fails to refine the solution after several rounds. Also, Lyra utilizes “proof -” to control the model response to be formal proof. Please refer to the more detailed answer to  **The difference between Lyra and other works** in **Author Response to All Reviewers**.
> >
> >
> >
> >
> > **Q7: Would CC work with the error messages from a different ITP?**
> >
> > A7: Yes, the CC can work with the error message from an ITP.
> >
> > The following is the requirement of Conjecture Correction.
> > * The pair of (input, output). For DSP [1], Subgoa-Learning [2] and Lyra, the input is (problem, informal proof), and the output is (formal proof).
> > * The feedback from the ITP. Given formal proof, the ITP should be able to validate and give feedback, such as error messages.
> > * The refinement instruction. The refine instruction is employed to ask the LLM to refine the previous response. The refinement instruction is shown in Algorithm 2.
> >
> > Moreover, we emphasize that Conjecture Correction can be an easy framework that can be extended to other tasks. It refines response with instruction and error messages but does not need several pairs of (input&error, output). Our work follows the previous experiment setting, such as DSP [1] and Subgoal-Learning [2], validating the proposed methods with Isabelle Prover on the miniF2F dataset.
> >
> > **Q8: How much does the availability of formal proofs and error messages for a particular ITP affect the effectiveness of CC?**
> >
> > A8: Excluding the first round, there is always a formal proof and the corresponding error message from a particular ITP. The proposed Conjecture Correction needs both previous formal proof and error messages from a particular ITP.
> >
> > The Conjecture Correction process is the following:
> > * In the first round, LLM generates a formal proof (just like DSP [1], without an error message), and then sends it for prover validation to get error messages.
> > * In the following round, Conjecture Correction combines previous formal proof and error message to generate a better formal proof and then sends it for prover validation to get error messages. Conjecture Correction repeats the process until one formal proof passes the prover validation.
> > * Therefore, there always exists a formal proof and an error message, excluding the first round. And Conjecture Correction needs both a formal proof and an error message.
> >
> > **Q9: How much does the quality of the error message affect CC?**
> >
> > A9: Raw error message from Isabelle Prover is enough for Conjecture Correction.
> >
> > In our experiment setting, we directly utilize the raw error message from Prover. The experiments have shown that raw error message is enough to improve performance. Future work may discuss how to post-process the raw error message from the prover for LLM to better understand.
> >
> > **Q10: Spelling errors in Algorithm 2.**
> >
> > A10: Thank you very much for your notice. We have fixed the spelling errors in Algorithm 2.
> >
> > If you think our comment and update address your concerns, could you please consider raising your rating of the paper? Thank you!
> >
> > [1] Jiang, A. Q., Welleck, S., Zhou, J. P., Li, W., Liu, J., Jamnik, M., ... & Lample, G. (2022). Draft, sketch, and prove: Guiding formal theorem provers with informal proofs. arXiv preprint arXiv:2210.12283.
> > [2] Zhao, X., Li, W., & Kong, L. (2023). Decomposing the Enigma: Subgoal-based Demonstration Learning for Formal Theorem Proving. arXiv preprint arXiv:2305.16366.

---

> > > ### Comment · Reviewer_NmsF · 2023-11-14
> > >
> > > I thank the authors for their long and detailed response. However, there seems to be a misunderstanding about some of my questions and I will try to clarify them below. I will refer to the question numbering used by the authors in their response.
> > >
> > > **Q2, Q3, Q4**: I understand that TC as a *technique* depends only on the ITP and not on the dataset. However, whether TC will be effective or not *does* depend on the dataset. There is no guarantee that performance gains due to TC demonstrated on the miniF2F dataset will also transfer to another dataset. Therefore, I fear that the heuristics presented in the paper might be overfit to improving the performance on the miniF2F dataset.
> > >
> > > **Q7, Q8**: Again, I understand that CC is a general technique and can work with other ITPs. However, the interesting empirical observation of this paper is that off-the-shelf LLMs (**without any fine-tuning**) are able to understand the Isabelle error messages and fix the proofs.  Would the LLMs evaluated in the paper be able to understand error messages from other ITPs? Do the authors have any hypotheses for why the LLMs are even able to understand Isabelle error messages (for instance, does the publicly available Isabelle proof corpus also have examples of error messages)?

---

> > > > ### Author Response · Authors · 2023-11-16
> > > > **Response to Reviewer NmsF (Part 1/2)**
> > > >
> > > > Dea Reviewer NmsF,
> > > >
> > > > Thank you very much for your reply. We will address your concerns below.
> > > >
> > > > **Q1: There is no guarantee that performance gains due to TC demonstrated on the miniF2F dataset will also transfer to another dataset**
> > > >
> > > > A1: There may be a misunderstanding here. In fact, it is guaranteed that performance gains due to TC are larger or equal to zero, for any dataset, compared to baseline sledgehammer+heuristics (which is w/o Tool Correction). We explain it in detail below.
> > > > * **Conjecture**. A conjecture is a conclusion or a proposition that is proffered on a tentative basis without proof. For example, a conjecture can be “x = 19∗(x div 19)+4”.
> > > > * **Automated Proving Tools**. Automated proving tools are employed to prove conjectures. For example, Isabelle has proving tools “by simp, by fastforce”, “sledgehammer” and so on. And also, we can add rules/lemma to corresponding proving tools to make them better, such as “ by (simp add: div mult mod eq)”.
> > > > * If all automated proving tools (such as sledgehammer, simp, and so on) fail to prove a conjecture, then the prover (such as Isabelle or Lean) fails to prove the conjecture and validate the whole formal proof. On the other hand, as long as one automated proving tool can prove the conjecture, then the prover successfully proves the conjecture, and the prover can continue to validate the whole formal proof.
> > > > * When failing to validate a conjecture of a formal proof, we fail to validate this whole formal proof. There are two situations in which the prover (such as Isabelle or Lean) fails to prove a conjecture.
> > > >    * The Conjecture is incorrect. If a conjecture is incorrect, then whatever powerful proving tools there are, we can not prove the conjecture, and so we fail to prove the whole formal proof.
> > > >    * The Conjecture is correct, but the proving tool is incorrect. For this situation, we can try to employ more powerful tools and try to prove the conjecture.
> > > >    * When we fail to validate a formal proof, we do not know whether the conjecture is incorrect, or the proving tool is incorrect. However, we can reduce the occurrence of “The Conjecture is correct, but the proving tool is incorrect”. Apparently, the better we can reduce the occurrence of this situation, the higher the pass rate.
> > > > * Compared to sledgehammer+heuristics, the proposed Tool Correction proposes to design more powerful rules to reduce the occurrence of “The Conjecture is correct, but the proving tool is incorrect”, and the sledgehammer+heuristics implementation is a subset of Tool Correction.
> > > >    * The implementation of sledgehammer + heuristics (implemented by DSP): given a tactic, if the tactic is the sledgehammer, then utilize sledgehammer + 11 tactics to close the conjecture.
> > > >    * The implementation of Tool Correction:  1) given a tactic, we first try the given tactic (whatever it is); 2) If it fails and it begins with "by" or "." or “sledgehammer”, then turn to all potential tools, such as a sledgehammer, simp, and so on.  For example, if “by (simp add: div mult mod eq)” fails, the DSP will immediately stop validating, while our proposed Tool Correction will try all other potential tools.
> > > >    * Therefore, given a conjecture, if sledgehammer+heuristics can prove it, then our proposed Tool Correction can prove it. And if sledgehammer+heuristics fails to prove it, our proposed Tool Correction still has the potential to prove it.
> > > > * Therefore, it is guaranteed that performance gains due to TC are larger or equal to zero, for any dataset, compared to baseline sledgehammer+heuristics (which is w/o Tool Correction).
> > > >
> > > >
> > > >
> > > >
> > > > **Q2: Would the LLMs evaluated in the paper be able to understand error messages from other ITPs?**
> > > >
> > > > A2: Recent work suggests that the LLMs can understand the error messages from other ITPS (such as Lean).
> > > >
> > > > Uploaded to Arxiv in October 2023, COPRA [1], utilizing the LLMs, interacts with Lean, showing that interaction with the environment helps in fixing the errors encountered while writing long proofs. The COPRA generates formal Lean proof tactic-by-tactic, and it utilizes the Lean error to fix the incorrect tactics. COPRA shows a successful fix example in its Paper Figure 4 on Page 5.
> > > > * According to the Figure, the initial tactic is “rw h”, which is wrong. The error message: “Got error in 'rw h’: error: rewrite tactic failed, did not find instance of the pattern in the target expression x%2.”
> > > > *  Then, the tactic becomes “apply nat.mul_mod_right” (also incorrect). The error message: “Got error in 'apply nat.mul mod right': error: invalid apply tactic, failed to unify x*x%2=0 with ?m_1* ?m_2 % ?m_1= 0”
> > > > *  And finally is the correct one “rw nat.mul_mod”.
> > > > * This suggests that the LLMs evaluated in the paper can understand error messages from other ITPs, such as Lean.

---

> ### Author Response · Authors · 2023-11-16
> **Response to Reviewer NmsF (Part 2/2)**
>
> **Q3: Do the authors have any hypotheses for why the LLMs are even able to understand Isabelle error messages (for instance, does the publicly available Isabelle proof corpus also have examples of error messages)**
>
> A3: There are two hypotheses for why the LLMs are even able to understand Isabelle error messages.
>
> Let’s start from an easy point: the LLMs can understand English and Isabelle syntax.
> * LLMs can understand English. This is approved by OpenAI. Therefore, this conclusion is obvious.
> * LLMs can understand Isabelle syntex.
>    * There are publicly available Isabelle proof corpus. For example, The entire AFP library, the
> largest formal library that contains most of Isabelle proofs, is 180MB in size [2].
>    * LLMs can understand Isabelle's syntax, and this is proved by previous works [2-5].
>
> Hypothesis 1: The publicly available Isabelle proof corpus also has examples of error messages.
> * The training dataset of GPT-3 [6] contains Common Crawl datasets [7], which contain Stackflow.
> * On the Stackflow, we find examples of error messages.
>    * Example 1: Failed to apply initial proof method: using this: [ ] ∈ ns_public goal (1 subgoal): 1. ∀A B X. Says A B X ∉ set_of_list [ ]
>    * Example 2: Failed to apply proof method: using this: (y, x) ∈ r^* (z, y) ∈ r goal (1 subgoal): 1. (z, x) ∈ r^*
>    * Example 3: Failed to apply initial proof method: using this: n < a n < b goal (1 subgoal): 1. n * n < a * b
> * Therefore, the publicly available Isabelle proof corpus may also have examples of error messages.
>
>
> Hypothesis 2: the LLMs may understand Isabelle's error message if they understand English and Isabelle's syntax, but do not have to see Isabelle's error messages before.
> * First, we show what Isabelle's error messages look like. The error message is written in English, such as “Failed to apply proof method using this:  0 < y goal (1 subgoal): 1. 9 * (x * sin x) + 4 / (x * sin x) =    (9 * (x * sin x)\<^sup>2 + 4) / (x * sin x) At command “by” “.
> * According to the error message, we can find that the error message only contains English words and Isabelle syntax (such as \<^sup>).
> * Therefore, to understand Isabelle's error message, the proposed LLM may not need to see Isabelle's error message before, but just has to understand English and Isabelle's syntax.
>
> If you think our comment and update address your concerns, could you please consider raising your rating of the paper? Thank you!
>
> [1] Thakur, A., Wen, Y., & Chaudhuri, S. (2023). A Language-Agent Approach to Formal Theorem-Proving. arXiv preprint arXiv:2310.04353.
>
> [2] Wu, Y., Jiang, A. Q., Li, W., Rabe, M., Staats, C., Jamnik, M., & Szegedy, C. (2022). Autoformalization with large language models. Advances in Neural Information Processing Systems, 35, 32353-32368.
>
> [3] Jiang, A. Q., Li, W., Tworkowski, S., Czechowski, K., Odrzygóźdź, T., Miłoś, P., ... & Jamnik, M. (2022). Thor: Wielding hammers to integrate language models and automated theorem provers. Advances in Neural Information Processing Systems, 35, 8360-8373.
>
> [4] Jiang, A. Q., Welleck, S., Zhou, J. P., Li, W., Liu, J., Jamnik, M., ... & Lample, G. (2022). Draft, sketch, and prove: Guiding formal theorem provers with informal proofs. arXiv preprint arXiv:2210.12283.
>
> [5] Zhao, X., Li, W., & Kong, L. (2023). Decomposing the Enigma: Subgoal-based Demonstration Learning for Formal Theorem Proving. arXiv preprint arXiv:2305.16366.
>
> [6] Brown, T., Mann, B., Ryder, N., Subbiah, M., Kaplan, J. D., Dhariwal, P., ... & Amodei, D. (2020). Language models are few-shot learners. Advances in neural information processing systems, 33, 1877-1901.
>
> [7] Raffel, C., Shazeer, N., Roberts, A., Lee, K., Narang, S., Matena, M., ... & Liu, P. J. (2020). Exploring the limits of transfer learning with a unified text-to-text transformer. The Journal of Machine Learning Research, 21(1), 5485-5551.

---

> > ### Comment · Reviewer_NmsF · 2023-11-16
> >
> > I again thank the authors for their long and detailed response.
> >
> > Regarding **Q1**, I understand that TC is guaranteed to not cause any reduction in performance. As the authors say, it is guaranteed that performance gains due to TC are larger or equal to zero. However, my point was simply that there is no guarantee that the performance gains due to TC will be larger than zero for any dataset. The reason TC helps for miniF2F might be because the heuristics in TC are tailored to this dataset. Is there a reason to believe that these heuristics will also be useful for other datasets?
> >
> > Thank you for the responses to **Q2** and **Q3**.  I think it would be helpful incorporate the discussion on why LLMs might be able to understand error messages in the paper.
> >
> > Overall, I remain unconvinced by the technical contribution of TC but I find the empirical phenomenon of LLMs being able to interpret ITP error messages interesting. I am updating my score to 5.

---

> ### Author Response · Authors · 2023-11-18
> **Response to Reviewer NmsF**
>
> Dear Reviewer NmsF,
>
> Thank you very much for the reply, and thank you very much for updating the score.
>
> We have incorporated the discussion on why LLMs might be able to understand error messages in the paper (please check the updated version). We will explain the question of TC in detail below.
>
> **Q1: However, my point was simply that there is no guarantee that the performance gains due to TC will be larger than zero for any dataset.**
>
> A1: **Our claim and this point does not have conflict**. There is a guarantee that TC does not cause any reduction in performance, and there is no guarantee that the performance gains due to TC will be larger than zero for any dataset. And we prove that **there does not exist** that “there is a guarantee that the performance gains due to Algorithm A will be larger than zero, compared to baseline sledgehammer+heuristics, for any dataset (**the pass rate limit is 100%**), for any Algorithm A”. We will explain it below.
>
> * **Recall when TC works**. As described in **previous Q1**, TC is employed to reduce the occurrence of “The Conjecture is correct, but the proving tool is incorrect”.
>    * For a very easy dataset, if any automated proving tool can prove the formal proof from the dataset, then TC may not further improve performance.
>    * For a very difficult dataset, if all conjectures are incorrect for the proof from the dataset, then TC may not further improve performance.
>    * For other datasets (have the occurrence of “The Conjecture is correct, but the proving tool is incorrect”), the proposed TC has the potential to further improve the performance.
> * Therefore, **both the following two claims are correct**:
>    * There is a guarantee that the performance gains due to TC will be larger than zero or equal to zero.
>    * There is no guarantee that the performance gains due to TC will be larger than zero.
>
> Meanwhile, as we both agree that “there is a guarantee that the performance gains due to TC will be larger or equal to zero, for any dataset”, we want to prove that there **does not exist** “there is a guarantee that the performance gains due to TC will be larger than zero, for any dataset”.
> * We already know that both TC and sledgehammer+heuristics (baseline) can solve some problems.
> * Then, we can select a subset, so that the subset’s problem can be solved by both TC and sledgehammer+heuristics.
>    * Therefore, the pass rate of sledgehammer + heuristics is 100%.
>    * If “there is a guarantee that the performance gains due to TC will be larger than zero, for any dataset”, then the pass rate of TC will be larger than 100%, which is a contradiction.
> * Therefore, there **does not exist** “there is a guarantee that the performance gains due to TC will be larger than zero, for any dataset”.
> * Actually, it is also easy to prove that there **does not exist** that “there is a guarantee that the performance gains due to Algorithm A will be larger than zero, compared to baseline sledgehammer+heuristics, for any dataset(**the pass rate limit is 100%**), for any Algorithm A”.
> * Hence, it is **good enough** that “there is a guarantee that the performance gains due to TC will be larger or equal to zero, for any dataset.”
>
>
> **Q2: The reason TC helps for miniF2F might be because the heuristics in TC are tailored to this dataset.**
>
> A2: We do not tailor the heuristics for the miniF2F dataset.
>
> These heuristics tools (such as sledgehammer, by simp and so on) are mentioned by the DSP paper (ICLR 2023, Oral). To have a fair comparison, we directly use and keep the same heuristics tools from DSP. Hence, we do not optimize the heuristics tools set for the miniF2F dataset, but directly use these mentioned heuristics tools from the DSP paper.
>
> Though TC and sledgehmmaer+heuristics employ the same heuristics, we both agree that there is a guarantee that TC is better than sledgehammer+heuristics (larger or equal to zero performance gains). Therefore, compared to baseline DSP’s sledgehammer+ heuristics, the improvement of TC is not caused by tailoring the heuristics for the miniF2F dataset.
>
> **Q3: Is there a reason to believe that these heuristics will also be useful for other datasets?**
>
> A3: There are two reasons to believe that these heuristics will also be useful for other datasets:
> * For any given formal proof (dataset), the TC always gets non-negative performance gains, compared with baseline sledgehammer +heuristics (we both agree).
>    * **That TC is helpful and good enough**. Because there does not exist that Algorithm A always gets positive performance gains for any dataset (**the pass rate limit is 100%**), compared to baseline sledgehammer+heuristics, for any Algorithm A (proved in Q1).
> * We do not tailor or optimize the heuristics set for the miniF2F dataset (proved in Q2).
>
>
> If you think our comment and update address your concerns, could you please consider raising your rating of the paper? Thank you!

---

> > ### Comment · Reviewer_NmsF · 2023-11-19
> >
> > I thank the authors for engaging in this discussion. My concerns about the general applicability of TC on datasets other than miniF2F remain and I will keep my updated score.

---

> ### Author Response · Authors · 2023-11-20
> **Response to Reviewer NmsF**
>
> Dear Reviewer NmsF,
>
> Thank you very much for your reply. We will address the concerns of TC's general applicability in the following.
>
> **Empirical Results**
> * **Experiment Setting**. Our experiment setting follows previous work, including DSP (ICLR 2023) and Sub-goal learning, which all only conduct experiments on the miniF2F dataset. To the best of our knowledge, the miniF2F is the only dataset that has both informal problem, informal proof and formal statement. And all DSP, subgoal-learning and Lyra need informal problems for inference.
> * To better analyze the TC, we modify the miniF2F to build an IMO problem set (which has 40 IMO problems). The IMO problem set is much more challenging than common automated theorem proving problems, which can effectively prove the TC generalization. The results are the following.
>
> |  Dataset   | Human Informal Proof |  Lyra w/o Tool Correction  |  Lyra | Improvement |
> |  ---- |  :----:  | :----:  | :----:  |  :----:  |
> | IMO problem set | &cross; |2.5% |  2.5% | 0.0%|
> | IMO problem set | &check; |0.0% |  2.5% | 2.5%|
>
> This proves that: for the other datasets, such as IMO problem set, Tool Correction can still improve the performance. And the emperical results also prove the previous conclusion: the performance gains of TC are larger or equal to zero, compared to baseline sledgehammer+heuristics, for any dataset.
>
> **Theoretical Analysis**
>
> The following is the analysis of TC's general applicability.
> * The miniF2F is a common and widely used dataset which contains different-level problems and various problem types, so that it can sufficiently evaluate the method. And DSP and subgoal-learning also use the dataset for all their experiments.
> * The TC presents an important idea: post-process LLM response to further improve the performance, which can be widely used for other datasets or tasks.
> * **TC generalization**. **We both agree** that the performance gains of TC are **larger or equal to zero**, compared to baseline sledgehammer+heuristics, for any dataset (TC does not cause performance reduction).
> * **Reviewer NmsF’s requirement of generalization**. the performance gains of TC are **larger than zero**, compared to baseline sledgehammer+heuristics, for any dataset.
>
> * **Obvious point**: as proved in Q2 of Nov 18 response, there is **no model/algorithm that can fullfills** the Reviewer NmsF’s requirement of generalization.
>
> According to both **Empirical Results** and **Theoretical Analysis**, the concern of TC's general applicability is addressed.
>
> If you think our comment and update address your concerns, could you please consider raising your rating of the paper? Thank you!

---

> > ### Comment · Reviewer_NmsF · 2023-11-20
> >
> > I again thank the authors for the discussion but perhaps we have to agree to disagree at this stage . Some responses below:
> >
> > > Reviewer NmsF’s requirement of generalization. the performance gains of TC are larger than zero, compared to baseline sledgehammer+heuristics, for any dataset.
> >
> > No, I don't expect TC to perform better than sledgehammer+heuristics for **any** dataset. Perhaps the way I phrased this was unclear and my apologies for the same. In my earlier message when I said that, "there is no guarantee that the performance gains due to TC will be larger than zero for any dataset", **any** was meant to be interpreted as an existential quantifier and not as a universal quantifier. My comment was in response to the authors' argument that performance gains due to TC will be zero or larger than zero for any dataset, and it was meant to highlight the fact such a guarantee does not really suggest that TC is going to improve performance on other datasets. It only suggests that TC is not going to hurt performance.
> >
> >
> > > To better analyze the TC, we modify the miniF2F to build an IMO problem set (which has 40 IMO problems).
> >
> > Doesn't miniF2F also include IMO problems?

---

> > > ### Author Response · Authors · 2023-11-21
> > > **Response to Reviewer NmsF**
> > >
> > > Dear Reviewer NmsF,
> > >
> > > The reply is received with thanks. According to the reply, we feel that Reviewer NmsF would like to see more experiment result of TC, especially TC performance on other dataset.
> > >
> > > The following are the results of DT-Solver [1] with Tool Correction on the PISA dataset [2].
> > > * Why not apply Lyra on the PISA dataset? Similarly to the experiment setting of DSP and Sub-goal learning, the Lyra needs informal information, such as informal problem and informal proof. To the best of our knowledge, only miniF2F has a Isabelle dataset that has informal information, while PISA does not.
> > > * Why choose DT-Solver? We used to reproduce the results of DT-Solver so that the experiment can be finished before the rebuttal deadline.
> > > * Why choose the PISA dataset? It is used to evaluate previous famous works, such as Lisa [2] Thor [3] and Thor + expert iteration [4].
> > >
> > > |  Dataset | DT-Solver   | DT-Solver + sledgehammer+heuristics  | DT-Solver + Tool Correction|
> > > |  ---- |  ----  | ----  | ----  |
> > > | PISA  | 37.0% | 37.0% |55.2% |
> > >
> > > According to the experiment results, we can observe that
> > > * Tool Correction significantly improves the performance, from 37.0 to 55.2%. This proves that Tool Correction is also useful for other datasets, such as the PISA dataset.
> > > * The sledgehammer+heuristics does not work well for DT-Solver on the PISA dataset, because DT-Solver hardly uses sledgehammer tactics, while sledgehammer+heuristics only works when the given tactics is sledgehammer.
> > > * We believe the above experiment can prove that TC is also useful for other datasets but not limited to miniF2F, even for different models. Meanwhile, TC is a plug-in-plug-out module, which is easily integrated into other methods.
> > >
> > >
> > >
> > >
> > > **Q1: "there is no guarantee that the performance gains due to TC will be larger than zero for any dataset", **any** was meant to be interpreted as an existential quantifier and not as a universal quantifier. My comment was in response to the authors' argument that performance gains due to TC will be zero or larger than zero for any dataset, and it was meant to highlight the fact such a guarantee does not really suggest that TC is going to improve performance on other datasets. It only suggests that TC is not going to hurt performance.**
> > >
> > > A1: Thank you very much for your clarification. All the following claims are correct, compared to baseline sledgehammer+heuristics.
> > > * For every dataset, performance gains of TC are guaranteed to be larger or equal to zero.
> > > * There exist some datasets so that the performance gains of TC are larger than zero.
> > > * There exist some datasets so that the performance gains of TC are zero.
> > > * There is no dataset so that the performance gains of TC are negative.
> > >
> > > Moreover, we have conducted additional experiments on another dataset PISA, to prove that TC can still improve performance on other datasets (in the above response).
> > >
> > > **Q2: Doesn't miniF2F also include IMO problems?**
> > >
> > > A2: The miniF2F contains different-level problems, such as undergraduate level and IMO level. The IMO problem set selects all IMO-level problems (a total of 40 problems) from miniF2F for further analysis.
> > >
> > > We sincerely hope the above response can address the concerns of TC. If there is anything else we could do, please let us know. If you think our comment and update address your concerns, could you please consider raising your rating of the paper? Thank you!
> > >
> > > [1] Wang, H., Yuan, Y., Liu, Z., Shen, J., Yin, Y., Xiong, J., ... & Liang, X. (2023, July). Dt-solver: Automated theorem proving with dynamic-tree sampling guided by proof-level value function. In Proceedings of the 61st Annual Meeting of the Association for Computational Linguistics (Volume 1: Long Papers) (pp. 12632-12646).
> > >
> > > [2] Jiang, A. Q., Li, W., Han, J. M., & Wu, Y. (2021). LISA: Language models of ISAbelle proofs. In 6th Conference on Artificial Intelligence and Theorem Proving (pp. 378-392).
> > >
> > > [3] Jiang, A. Q., Li, W., Tworkowski, S., Czechowski, K., Odrzygóźdź, T., Miłoś, P., ... & Jamnik, M. (2022). Thor: Wielding hammers to integrate language models and automated theorem provers. Advances in Neural Information Processing Systems, 35, 8360-8373.
> > >
> > > [4] Wu, Y., Jiang, A. Q., Li, W., Rabe, M., Staats, C., Jamnik, M., & Szegedy, C. (2022). Autoformalization with large language models. Advances in Neural Information Processing Systems, 35, 32353-32368.

---

> > > > ### Comment · Reviewer_NmsF · 2023-11-21
> > > >
> > > > I thank the authors for all the hard work in running these additional experiments to address my concerns. These new results are impressive and I am increasing my score to 6. I hope that the authors will include these additional results and discussions in the paper (or its appendix).
> > > >
> > > > As a non-expert, there is one question that comes to mind. Why does the simple strategy used by TC, i.e., iteratively replacing the tactics in a failed proof with one of the 11 tactics from the *tool_heuristics* set, work? Does it tell us something fundamental about the nature of mathematical proofs? Or is it an artifact of the types of problems included in datasets such as miniF2F and PISA? I would guess that the same set of 11 tactics are not effective for proofs from every branch of mathematics.
> > > > I do not expect the authors to answer these questions. Instead, I present these as questions that came to my mind as a reader. However, if the authors do have thoughts about this, it would help the paper to include a discussion on it.

---

> > > > > ### Author Response · Authors · 2023-11-21
> > > > > **Response to Reviewer NmsF**
> > > > >
> > > > > Dear Reviewer NmsF,
> > > > >
> > > > > The reply is received with thanks! Thank you very much for improving the score. Thank you very much for your engagement in the discussion. We will explain the questions below and add them to our updated version.
> > > > >
> > > > > **Q1: Why does the simple strategy used by TC, i.e., iteratively replacing the tactics in a failed proof with one of the 11 tactics from the tool_heuristics set, work?**
> > > > >
> > > > > A1: Reason: TC reduces the occurrence of “the conjecture is correct, but the proving tools fail to prove the conjecture”.
> > > > > * Recall when can we successfully prove a conjecture
> > > > >    * The conjecture should be correct, and the proving tools (such as a sledgehammer) can prove the conjecture.
> > > > > * Recall when we fail to prove a conjecture: the proving tools fail to prove the conjecture.
> > > > >    * Situation 1: The conjecture is incorrect
> > > > >    * Situation 2: Or, the correction is correct, but the proving tool is not correct so the proving tool cannot prove the conjecture.
> > > > > * **Observation**: prover (such as Isabelle or Lean) fails to validate a formal proof, which may caused by **“the conjecture is correct, but the proving tools fail to prove the conjecture”**. The following is an example.
> > > > >    * Conjecture: 1*1=1
> > > > >    * If the proving tool can not process multiplication operation, then it will fail to prove the conjecture 1*1=1, though the conjecture is correct.
> > > > > * Therefore, to improve the performance, we can **reduce the occurrence of “the conjecture is correct, but the proving tools fail to prove the conjecture”**
> > > > >    * **One solution is Tool Correction**: try as many proving tools as we can to prove the conjecture.
> > > > >
> > > > > **Q2: Does it tell us something fundamental about the nature of mathematical proofs? Or is it an artifact of the types of problems included in datasets such as miniF2F and PISA?**
> > > > >
> > > > > A2: Yes, this tells us something fundamental about the nature of mathematical proofs: **try as many proving tools as we can, so that we can improve the performance by reducing the occurrence of “the conjecture is correct, but the proving tools fail to prove the conjecture”.**
> > > > >
> > > > >
> > > > > **Q3: I would guess that the same set of 11 tactics are not effective for proofs from every branch of mathematics**
> > > > >
> > > > > A3:  To improve the effectiveness of TC, we can easily add tactics, including by clarify, by fact, by rule, by erule, by elim, by induction, by algebra and so on.
> > > > >
> > > > > According to the DSP paper, the 11 tactics can be used to process algebra and number theory problems. And the tactics also have their own advantages and disadvantages. For example, it seems that “by simp” is not very good at processing natural numbers, as shown in Figure 1 in our paper.
> > > > >
> > > > > As introduced in the DSP paper, the Sledgehammer [1] is relatively special, as it works by flattening the goals encoded in the higher-order logic used by Isabelle/HOL into other logics (e.g., first-order logic) which can then be fed into automated theorem provers.  The development of proving tools is also an important direction of automated theorem proving. Recently, there has been a work named Magnushammer [2], which is more powerful than sledgehammer.
> > > > >
> > > > > We promise that there will be a part of the discussion about these questions in our updated version.
> > > > >
> > > > > **If there is any question or problem, please let us know. We will try our best to answer any other problems!** We sincerely hope that this discussion is also helpful for Reviewer NmsF.
> > > > >
> > > > > [1]  Paulson, L., & Blanchette, J. Three years of experience with sledgehammer, a practical link between automatic and interactive theorem provers (2015). DOI: https://doi. org/10.29007/tnfd.
> > > > >
> > > > > [2] Mikuła, M., Antoniak, S., Tworkowski, S., Jiang, A. Q., Zhou, J. P., Szegedy, C., ... & Wu, Y. (2023). Magnushammer: A transformer-based approach to premise selection. arXiv preprint arXiv:2303.04488.

---

### Official Review · Reviewer_1Jjk · 2023-10-31

**Soundness:** 3 good
**Presentation:** 3 good
**Contribution:** 3 good
**Rating:** 6
**Confidence:** 4

**Summary:**

This paper presents an LLM-augmented automated theorem proving framework called Lyra in a formal theorem proving environment. The distinguishing features of Lyra include Tool Correction (TC) and Conjecture Correction (CC), which respectively post-edit formal proofs emitted from LLMs given feedback from the proving environment. Good performance has been shown over the miniF2F dataset with 3 IMO problems being solved.

**Strengths:**

The paper is well written with a clear illustration of its two contributions TC and CC. I especially appreciate the ablation study where Lyra downgrades to DSP without TC and CC. The performance gain also looks good.

**Weaknesses:**

Although I very much like the idea of CC, the innovation of TC appears slightly limited. At least to me, it does not involve much interaction with LLMs -- it is more like an exhaustive attempt on a set of heuristically chosen proof methods other than Sledgehammer.

**Questions:**

- page 1, 'However, they have not been able to post-process LLM generation or gradually refine previous generations.': The Baldur paper (https://arxiv.org/abs/2303.04910) has explored post-processing LLM-generated proofs. I would love to see a comparison between Baldur and Lyra if possible.
- This paper is mostly based on the previous DSP paper, where LLMs are used to produce proof skeletons (i.e., unproved conjectures). In the tool correction part, it appears that LLMs are prompted to produce a full mechanised proof including the tactic 'by (simp add: div mult mod eq)', which is considered as LLM hallucination by the authors. Is that the case? If so, it might be a good idea to make the distinction clear as this may affect the ablation study.
- Figure 3, the informal proof and the formal sketch are actually quite different. For example, the formal one does not cover continuity nor limit, which have been mentioned several times in the informal proofs. I was wondering if the authors could elaborate a bit on the discrepancy between the informal proof and the formal one.

minor
- page 3, 'conducted on LLLMs' -> 'conducted on LLMs'
- page 5, 'As all formal proof begins with proof -': strictly speaking this is not quite true, as some Isabelle proofs start with 'proof (...)' or 'apply (...)', where ... can be some Isabelle tactics.

---

> ### Author Response · Authors · 2023-11-14
> **Response to Reviewer 1Jjk (Part 1/2)**
>
> Dear Reviewer 1Jjk,
>
> Thank you for appreciating our approach. We address your comments below.
>
> **Q1: the innovation of TC appears slightly limited.**
>
> A1: TC implementation reveals an important point: post-process of formal proof can significantly further improve performance, which does not get enough attention in previous work. Future work can explore how to design better post-process rules or techniques to improve the formal proof quality. And this is one important point that we want to present and prove via Tool Correction. For more details, please refer to **The difference between Tool Correction and sledgehammer+heuristic** in **Author Response to All Reviewers**.
>
> **Q2: I would love to see a comparison between Baldur and Lyra if possible.**
>
> A2: The following is our comparison between Baldur and Lyra. We have clarified this in our general response to all reviewers. Please kindly check our answer in **The difference between Lyra and other works** in **Author Response to All Reviewers**.
>
> |  Method   | Training-Free  |Information Besides Response and Error  |Reset Initial solution| Control Indicator |
> |  ----  | ----  | ----  | ----  | ----  |
> | Baldur  |&cross;  |&cross;  |&cross;  |&cross;  |
> | Lyra  | &check; |&check; |&check; |&check; |
>
> Compared to Baldur which needs a training process, the proposed Lyra is training-free. As Baldur does not release code or conduct experiments on miniF2F, currently we cannot directly compare their performance. However, we can compare Baldur, Thor [1] and Lyra to draw a conclusion between Baldur and Lyra.
>
> |  Method   | AFP-Computer Science  | AFP-Logic  | AFP-Mathematics  | AFP-Tools|  miniF2F-valid|miniF2F-test|
> |  ----  | ----  | ----  | ----  | ----  | ----  | ----  |
> | Baldur  | 50.0%  | 51.6%  | 41.9%  |53.9%  |-  |-  |
> | Thor [1]  |57.5% | 53.6% |  50.5%  |51.8%  |28.3%  |29.9%  |
> | Lyra  |- | - | -  |-  |55.3%  |51.2%  |
>
> Fortunately, we can 1) first compare Baldur with Thor [1] on AFP topic classification; 2) then compare Thor with Lyra on miniF2F; 3) finally, conclude: that Lyra is significantly better than Thor, while Thor is better or comparable to Baldur.
>
>
>
> **Q3: In the tool correction part, it appears that LLMs are prompted to produce a full mechanised proof including the tactic 'by (simp add: div mult mod eq)', which is considered as LLM hallucination by the authors. Is that the case? If so, it might be a good idea to make the distinction clear as this may affect the ablation study.**
>
> A3: Yes, that is the case. And we update the following to make the distinction clear.
>
> * Page 1: “ As shown in the observation in Figure 1, prover fails to prove conjecture x = 19 ∗ (x div 19) + 4 because LLM wrongly believes that by (simp add: div mult mod eq) can prove x = 19 ∗ (x div 19) + 4” —>  “ As shown in the observation in Figure 1, prover fails to prove conjecture x = 19 ∗ (x div 19) + 4 because by (simp add: div mult mod eq) generated by LLM cannot prove x = 19 ∗ (x div 19) + 4 (considered as LLM hallucination)”
> * Figure 1: “The prover fails because LLM wrongly believes that by (simp add: div mult mod eq) can prove x = 19∗(x div 19)+4” —> “The prover fails because by (simp add: div mult mod eq) generated by LLM cannot prove x = 19∗(x div 19)+4, which is considered as the hallucination.”
> * Moreover, we have highlighted that this is the LLM hallucination in our paper in Page 4: “For instance, consider the statement x = 19 ∗ (x div 19) + 4, where LLM proposes to utilize the tactic by (simp add: div mult mod eq), leading to failure. This is the LLM hallucination…”
>
>
> **Q4: Informal proof and the formal sketch are quite different. Elaborate a bit on the discrepancy between the informal proof and the formal one.**
>
> A4: The informal proof is a guide to formal proof, and it is not necessary that the informal proof steps and formal proof steps are one-to-one. The detailed explanation is the following.
> * Why Informal proof and the formal sketch are quite different? $Reason$: The informal proof is a guide to formal proof generation. According to the original DSP paper, the informal proof only needs to be useful for producing a sketch in the next stage.
> * Why the formal one does not cover continuity or limit? $Reason$: Actually, the formal one covers the limit. For example, the formal proof has shown that "ultimately have "1 < ?S" by simp [ATPWithTC]", which covers the limit. The limit is proved via the ATP and Tool Correction.

---

> > ### Author Response · Authors · 2023-11-14
> > **Response to Reviewer 1Jjk (Part 2/2)**
> >
> > **Q5: 'conducted on LLLMs' -> 'conducted on LLMs'**
> >
> > A5: Thank you very much for your notice. We have fixed the typos.
> >
> > **Q6: Some Isabelle proofs start with 'proof (...)' or 'apply (...)'**
> >
> > A6: Thank you very much for your notice. The following is the new description: As most formal proofs of our prompts begin with “proof -”, we add “proof -” at the end of the instruction so that the LLM response is formal proof.
> >
> > If you think our comment and update address your concerns, could you please consider raising your rating of the paper? Thank you!
> >
> > [1] Jiang, A. Q., Li, W., Tworkowski, S., Czechowski, K., Odrzygóźdź, T., Miłoś, P., ... & Jamnik, M. (2022). Thor: Wielding hammers to integrate language models and automated theorem provers. Advances in Neural Information Processing Systems, 35, 8360-8373.

---

> > > ### Comment · Reviewer_1Jjk · 2023-11-21
> > > **Thank you for the elaborate responses.**
> > >
> > > I sincerely thank the authors for the elaborate responses to my queries, most of which have been adequately addressed. Nevertheless, I still see the limitation of the current TC: it focuses on the 'by'-style proofs, which are essentially single-step proofs and leave little space for the LLMs to interact with the ITP. Most of those single step proofs fall within the scope of sledgehammer + heuristic tactics. Longer proofs starting with 'apply' could involve more interactions the ITP and tackle more complicate proofs beyond a single 'by'-style proof. In daily theorem proving, we often find a proof through a sequence of 'apply' steps and compress these steps into a single 'by' step in the cleanup phase; similar procedures also occur in Lean, which is termed as 'golf'. I see the authors plan to explore the 'apply'-style proofs, which could certainly be a valuable extension to the current work. For now, I will keep my score.

---

> ### Author Response · Authors · 2023-11-22
> **Response to Reviewer 1Jjk (Part 1/2)**
>
> Dear Reviewer 1Jjk,
>
> Thank you very much for your reply. The following is one of the potential extensions of TC for ‘apply'-style proofs. The difference between "by" and "apply" is in a pdf named "The Isabelle/Isar Reference Manual", while "by" is mainly introduced on Page 148 and "apply" is introduced on Page 167.
>
> * Recall the core idea of TC: replace potential incorrect tools/tactics with predefined correct ones. In this setting, we replace incorrect "apply"-style tactics with 12 predefined "apply"-style tactics.
> * Step 1:  Check whether we need to process the "apply" tactics. If we need and the previous tactic does not begin with "apply", then we directly change ‘apply'-style proofs to single-step method, which is sledgehammer+11 tactics. And if succeed via sledgehammer+11 tactics, then we continue. That is, if succeed via sledgehammer+11 tactics, for the next tactic, if it still begins with “apply”, we directly ignore it. If fails, we move to Step 2
> * Step 2: Try the given "apply"-style tactics. If fails, we move to Step 3.
> * Step 3: If Step 2 fails, then try sledgehammer+11 tactics. And if it succeeds, we continue. And if it fails, we try all 12 predefined "apply"-style tactics.
> * If Step 3 fails, then the prover fails.
>
>
>
>
> The following is the Python-like pseudocode.
>
> ```python
> #tactic_list: list of the tactics of formal proof
> #prover: Isabelle Prover
> #TCUsage: whether employ Tool Correction
> tool_heuristics=['by auto','by arith','by blast', 'by simp',
> 'by fastforce', 'by force', 'by eval', 'by presburger', 'by sos',
> 'by linarith', 'by (auto simp: field_simps)', 'sledgehammer']
>
>
> # can be other heuristics, such as dynamic heuristics.
> apply_heuristics=['apply auto','apply arith','apply blast', 'apply simp',
> 'apply fastforce', 'apply force', 'apply eval', 'apply presburger', 'apply sos',
> 'apply linarith', 'apply (auto simp: field_simps)', 'apply assumption']
>
>
> output={}
> previous_tactics="None"
> for tactic in tactic_list:
>     use_heuristics=False
>
>     if tactic.strip().startswith("done") and output.has_key('ignore_apply'): #solve "apply"-style via "by".
>         previous_tactics="done"
>         continue
>
>     # Step 1 Begin
>     if tactic.strip().startswith("apply") and TCUsage:
>         if output.has_key('ignore_apply') and previous_tactics.startswith("apply"):
>             # solved by previous tactics, such as single-step
>             continue
>         elif not (previous_tactics.startswith("apply")) #If previous one is not "apply"
>             for tool_try in tool_heuristics: # try single-step
>                 output = prover.run_tac(tool_try)
>                 if output['error'] is None:
>                     break
>             if output['error'] is None:
>                 #go to next tactic, and ignore the following "apply"
>                 #becuase already finish the proof
>                 output['ignore_apply']=True #solve the proof
>                 continue
>         else:
>             #do nothing
>     # Step 1 End
>
>
>     previous_tactics=tactic
>
>     #Step 2 Begin
>     output = prover.run_tac(tactic)
>     #Step 2 End
>
>     if output['error'] is not None:
>         if TCUsage: # Use Tool Correction or Not
>             if tactic.strip().startswith("by") or tactic.strip()==("."):
>                 use_heuristic=True
>
>         if ("sledgehammer" in tactic) or use_heuristic:
>             for tool_try in tool_heuristics:
>                 output = prover.run_tac(tool_try)
>                 if output['error'] is None:
>                     output['ignore_apply']=True
>                     break
>
>
>         # Step 3 Begin
>         if TCUsage and tactic.strip().startswith("apply"):
>             for tool_try in tool_heuristics:
>                 output = prover.run_tac(tool_try)
>                 if output['error'] is None:
>                     output['ignore_apply']=True #solve the proof
>                     break
>
>             if output['error'] is not None:
>                 for apply_try in apply_heuristics:
>                     output = prover.run_tac(apply_try)
>                     if output['error'] is None:
>                         # if not use by-style tactic,
>                         # not sure whether the whole proof is solved or not here.
>                         # So that NO output['ignore_apply']=True
>                         break
>         # Step 3 End
>
>
>     if output['error'] is not None:
>         return "tactic_failed", output
>     if output['tactic_state'] == 'no goals':
>         return "success", output
>
> return "proof_incomplete", output
> ```

---

> ### Author Response · Authors · 2023-11-22
> **Response to Reviewer 1Jjk (Part 2/2)**
>
> Another direction could be dynamic heuristics. For example, though the given tactic is incorrect (such as the "by (simp add: div_ mult_mod_eq) in Figure 1), there is still some useful information (such as something like div_mult_mod_eq). We may design corresponding methods to utilize the information. For example, search the lemma that is similar to div_mult_mod_eq and replace it. Therefore, another direction of TC extension could be dynamic tactic creation.
>
> If you think our comment and update address your concerns, could you please consider raising your rating of the paper? Thank you!

---

> > ### Comment · Reviewer_1Jjk · 2023-11-23
> >
> > I believe the dynamic tactic creation would be a more appealing direction of TC in the future. Note that there has been a static/pre-defined tactic exploration framework in Isabelle [1].
> >
> > [1] Nagashima, Yutaka, and Ramana Kumar. "A proof strategy language and proof script generation for Isabelle/HOL." Automated Deduction–CADE 26: 26th International Conference on Automated Deduction, Gothenburg, Sweden, August 6–11, 2017, Proceedings. Springer International Publishing, 2017.

---

> > > ### Author Response · Authors · 2023-11-23
> > > **Response to Reviewer 1Jjk**
> > >
> > > Dear Reviewer 1Jjk,
> > >
> > > Thank you very much for your reply. Thank you very much for your support for the direction of dynamic tactic creation. And thank you for your recommendation for the static/pre-defined tactic exploration framework PSL.
> > >
> > > Again, thank you very much for your attention and support to this work, and wish you a good day.
> > >
> > > Best regards, Paper 1096 Authors

---

### Official Review · Reviewer_Wc5t · 2023-11-03

**Soundness:** 4 excellent
**Presentation:** 4 excellent
**Contribution:** 2 fair
**Rating:** 6
**Confidence:** 5

**Summary:**

This paper proposes two methods for postprocessing and fixing the errors in the proof steps generated by LLM for theorem proving. The first method is Tool Correction, which tries a list of automation tactic such as sledgehammer, auto and arith on the fails steps. The second method is Conjecture Correction, which asks LLM to regenerate the proof step based on the error message or simply regenerate this step to start a new iteration. Experiments show that TC and CC could improve the performance significantly on the miniF2F valid (50.4%->55.3%) and test set( 42.6%->51.2%), and achieves the new state-of-the-art results.

==================================
Post-rebuttal:
After reading authors' responses and other reviews, I upgraded my score to 6.
Authors' response largely addressed my questions. I still think the technical novelty of this paper is relatively weak. But the extensive experiments and ablation studies (including the new experiment results presented in the rebuttal) are very solid and could be helpful to the AITP community.

**Strengths:**

Both Tool Correction and Conjecture Correction are technically sound. Tool Correction implies the insights that LLM is better at generating the next conjecture to prove than closing the conjecture. Extensive search is helpful to close the conjecture.

**Weaknesses:**

The paper doesn't have much novelty in terms of the approach. It seems that the set of 11 tactics in TC have been proposed in DSP. The method of appending error message for self-debugging and generating multiple candidates have been commonly used for code generation.

**Questions:**

1 I think one good baseline would be replace Minerva and Codex in DSP with GPT4. So we still have GPT4 to generate the proof sketch (the intermediate conjectures) but use 11 tactics + sledgehammer to close the open goals.
2 Could you calculate the number of wrong proof steps fixed by each tactic in TC?
3 The proposed method adds a lot more computation. How much time would be token by calling GPT4, TC and CC? If we set a time limit for each question like 10 or 30 minutes, what would be the performance of TC/CC compared with the GPT4 baseline and DSP+GPT4 baseline?

**Details Of Ethics Concerns:**

No ethics concern.

---

> ### Author Response · Authors · 2023-11-14
> **Response to Reviewer Wc5t (Part 1/2)**
>
> Dear Reviewer Wc5t,
>
> Thank you for the detailed review. We will address your concerns below.
>
> **Q1: novelty in terms of the approach.**
>
> A1: For Tool Correction, it provides this insight: post-process of formal proof can significantly further improve performance. Please refer to the **The difference between Tool Correction and sledgehammer+heuristic** in **Author Response to All Reviewers**.
>
> And, we also show the **The difference between Lyra and other works** in **Author Response to All Reviewers** to present the innovation of Conjecture Correction.
>
>
> **Q2: I think one good baseline would be replace Minerva and Codex in DSP with GPT4. So we still have GPT4 to generate the proof sketch (the intermediate conjectures) but use 11 tactics + sledgehammer to close the open goals.**
>
> A2: We have shown the result in Table 2 in our paper. Our proposed Lyra degrades to DSP if removes both Tool Correction and Conjecture Correction. That is, $DSP+Tool Correction + Conjecture Correction = Lyra$, while the original DSP has sledgehammer + 11 tactics  implementation (only works when the given tactic is "sledgehammer"), achieving 50.4% on miniF2F-valid and 42.6% on miniF2F-test.
>
> |  Method   | Model |TC  | CC  | miniF2F-valid|miniF2F-test|
> |  ----  | ----  | ----  | ----  | ----  | ----  |
> | DSP  | GPT-4|&cross; |&cross; |50.4% |42.6% |
> | DSP+TC  | GPT-4|&check; |&cross; |52.8% |45.9% |
> | DSP+CC  | GPT-4|&cross; |&check; |46.7% | 43.0% |
> | DSP+TC+CC (Lyra)  | GPT-4|&check; |&check; |55.3% |51.2% |
>
>
> **Q3: Could you calculate the number of wrong proof steps fixed by each tactic in TC?**
>
> A3: The following is the number of wrong proof steps fixed by each tactic in TC.
> * Proof comes from: the miniF2F validation set (pass rate 55.3%) and test set result (pass rate 51.2%), with GPT-4, human informal proof, Conjecture Correction and Tool Correction.
> * The definition of proof step: a proof step is regarded as a tactic.
> * Calculation protocol: if sledgehammer+heuristic or Tool Correction fails to validate the current tactic, then the current proving process will be terminated and we will turn to the next formal proof validation. Finally, we calculate how many correct tactics/proof steps.
>
> |  Dataset | Sledgehammer+heuristics   | Tool Correction  | Number of Fixed Steps|
> |  ---- |  ----  | ----  | ----  |
> | miniF2F-valid  |2260  | 3486 |1226 |
> | miniF2F-test  |2594  | 3887 |1293 |
>
> On miniF2F-valid, sledgehammer+heuristics can help the prover successfully pass 2260 steps. After adding Tool Correction, the number increases to 3486 steps. Therefore, Tool Correction fixes 1226 wrong steps. On miniF2F-test, Tool Correction fixes 1293 wrong steps
>
>
> **Q4: The proposed method adds a lot more computation. How much time would be token by calling GPT4, TC and CC?**
>
> A4: There may be a slight misunderstanding here. Our method running time is similar to DSP.
>
> Both tool Correction and Conjecture Correction do not add a lot more computing time.  The upper bound of Tool Correction time cost is 120+11*10=230 seconds for each tactic (same as the sledgehammer+heuristics in DSP), and formal proof generation (Calling GPT-4 and CC)  time varies from 10 seconds to 2 minutes. On average, Lyra takes about 2 minutes per attempt, including the time cost of TC, and GPT4+CC.
>
> For the Tool Correction, it only works when the current tactics fail and begin with "by" or "." or “sledgehammer”. And it has the same timeout upper bound as DSP's sledgehammer+hsuristics. For sledgehammer, the timeout is 120 seconds. For every of the other 11 tactics, the timeout is 10 seconds. Hence, the upper bound of Tool Correction is 120+11*10=230 seconds. Usually, the prover validation with TC can be finished in 1 minute.
>
> For the GPT-4+Conjecture Correction, it also does not add additional computing time. The proposed Lyra or DSP employs the aggressive model (Codex or GPT-4) to generate a formal proof. Hence, the time cost mainly depends on the length of the generated formal proof, which depends on different problems but not the Conjecture Correction. Depending on the difference of the problem, the formal proof generation is different for one attempt.
> * If the formal proof is short (such as an easy proof), formal proof generation only needs less than 10 seconds. For example, the correct formal proof is only "by sos".
> * If the formal proof is long (such as a difficult proof), formal proof generation needs several minutes. We have shown a long IMO formal proof example in Figure 3.
>
> Therefore, both Tool Correction and Conjecture Correction do not add a lot more computing time. The Lyra has the same time complexity as DSP.

---

> ### Author Response · Authors · 2023-11-14
> **Response to Reviewer Wc5t (Part 2/2)**
>
> **Q5: If we set a time limit for each question like 10 or 30 minutes, what would be the performance of TC/CC compared with the GPT4 baseline and DSP+GPT4 baseline**
>
> A5: The result depends on whether it allows the parallel process.
>
> **As there is no GPT4 baseline (for DSP paper, they also do not have experiment results of Codex/Minerva baseline, but have DSP+Codex and DSP+Minerva)**, we list the performance of Lyra and DSP+GPT4 in the following.
>
> If allows the parallel process, then we can keep the performance (55.3% on validation and 51.2% on test), if the time limit is larger or equal to 10 minutes. For the relationship between time limit and performance, we can refer to Figure 2 on Page 8, which presents the relationship between the number of attempts and the performance. One attempt takes 2 minutes, if not allow the parallel process.
>
> The following is the performance under different time limits for each question, for DSP(GPT-4) and Lyra.
>
> Parallel Process|  Time Limit   | Method  | Time Cost  | miniF2F-valid  | miniF2F-test|
> |  ----  | ----  | ----  | ----  | ----  |----  |
> &cross;| 10 mins  | DSP(GPT4)  | 10 mins  | 32.7%  |25.4%  |
> &cross;| 10 mins  | Lyra  | 10 mins  | 33.6%  |28.6%  |
> &cross;| 30 mins  | DSP(GPT4)   | 30 mins  | 40.1%  |31.5%  |
> &cross;| 30 mins  | Lyra  | 30 mins  | 40.1%  |36.0%  |
> &cross;| 400 mins  | DSP(GPT4)   | 400 mins  | 50.4%  |42.6%  |
> &cross;| 400 mins  | Lyra  | 400 mins |  55.3%  |51.2%  |
> &cross;| 600 mins  | DSP(GPT4)   | 400 mins  | 50.4%  |42.6%  |
> &cross;| 600 mins  | Lyra  | 400 mins |  55.3%  |51.2%  |
> &check;| 10 mins  | DSP(GPT4)   | 2 mins |  50.4%  |42.6%  |
> &check;| 10 mins  |Lyra | 10 mins |  55.3%  |51.2%  |
> &check;| 30 mins  | DSP(GPT4)   | 2 mins |  50.4%  |42.6%  |
> &check;| 30 mins  |Lyra | 10 mins |  55.3%  |51.2%  |
>
> Usually, it takes about 1～2 minutes to finish one attempt (we take 2 min/attempt here), where each problem is allowed to try 200 attempts(a total of 400 minutes), in our setting. For DSP, the 200 attempts can be processed in parallel. Hence, if allowing the parallel process, the maximum time cost is 2 minutes. For Lyra, these 200 attempts are divided into 40 patches, where each patch contains five attempts. The 40 patches can be processed in parallel.  Hence, if allowing the parallel process, the maximum time cost is 10 minutes. If not allowing the parallel process, when the time limit is 10 mins, Lyra achieves 33.6% miniF2F-validation and 28.6% miniF2F-test. And if not allow the parallel process, when the time limit is 30 mins, Lyra achieves 40.1% miniF2F-validation and 36.0% miniF2F-test.
>
> If you think our comment and update address your concerns, could you please consider raising your rating of the paper? Thank you!

---

> > ### Author Response · Authors · 2023-11-21
> > **Kind Reminder**
> >
> > Dear Reviewer Wc5t,
> >
> > Hope this finds you well. We sincerely hope that the above reply could address the concerns.
> >
> > As the discussion deadline is approaching (last 48 hours), if possible, could we know whether there are any additional concerns?
> >
> > Again, thank you very much for your attention to this work.
> >
> > Best regards, Paper 1096 Authors

---

> > > ### Author Response · Authors · 2023-11-22
> > > **Addtional Tool Correction Result**
> > >
> > > Dear Reviewer Wc5t,
> > >
> > > We conduct additional experiment results of TC to prove the generalization.
> > >
> > > The following are the results of DT-Solver [1] with Tool Correction on the PISA dataset [2].
> > > * Why not apply Lyra on the PISA dataset? Similarly to the experiment setting of DSP and Sub-goal learning, the Lyra needs informal information, such as informal problem and informal proof. To the best of our knowledge, only miniF2F has a Isabelle dataset that has informal information, while PISA does not.
> > > * Why choose DT-Solver? We used to reproduce the results of DT-Solver so that the experiment can be finished before the rebuttal deadline.
> > > * Why choose the PISA dataset? It is used to evaluate previous famous works, such as Lisa [2] Thor [3] and Thor + expert iteration [4].
> > >
> > > |  Dataset | DT-Solver   | DT-Solver + sledgehammer+heuristics  | DT-Solver + Tool Correction|
> > > |  ---- |  ----  | ----  | ----  |
> > > | PISA  | 37.0% | 37.0% |55.2% |
> > >
> > > According to the experiment results, we can observe that
> > > * Tool Correction significantly improves the performance, from 37.0 to 55.2%. This proves that Tool Correction is also useful for other datasets, such as the PISA dataset.
> > > * The sledgehammer+heuristics does not work well for DT-Solver on the PISA dataset, because DT-Solver hardly uses sledgehammer tactics, while sledgehammer+heuristics only works when the given tactics is sledgehammer.
> > > * We believe the above experiment can prove that TC is also useful for other datasets but is not limited to miniF2F, even for different models. Meanwhile, TC is a plug-in-plug-out module, which is easily integrated into other methods.
> > >
> > > [1] Wang, H., Yuan, Y., Liu, Z., Shen, J., Yin, Y., Xiong, J., ... & Liang, X. (2023, July). Dt-solver: Automated theorem proving with dynamic-tree sampling guided by proof-level value function. In Proceedings of the 61st Annual Meeting of the Association for Computational Linguistics (Volume 1: Long Papers) (pp. 12632-12646).
> > >
> > > [2] Jiang, A. Q., Li, W., Han, J. M., & Wu, Y. (2021). LISA: Language models of ISAbelle proofs. In 6th Conference on Artificial Intelligence and Theorem Proving (pp. 378-392).
> > >
> > > [3] Jiang, A. Q., Li, W., Tworkowski, S., Czechowski, K., Odrzygóźdź, T., Miłoś, P., ... & Jamnik, M. (2022). Thor: Wielding hammers to integrate language models and automated theorem provers. Advances in Neural Information Processing Systems, 35, 8360-8373.
> > >
> > > [4] Wu, Y., Jiang, A. Q., Li, W., Rabe, M., Staats, C., Jamnik, M., & Szegedy, C. (2022). Autoformalization with large language models. Advances in Neural Information Processing Systems, 35, 32353-32368.

---

> ### Author Response · Authors · 2023-11-23
> **Integrate Tool Correction with "apply"-style tactic**
>
> Dear Reviewer Wc5t,
>
> The proposed **Tool Correction can also be integrated with other Isabelle tactics besides "by"-style tactic** (single-step method), **while previous work (such as sledgehammer+heuristics in DSP) can not** (only work when the given tactic is "sledgehammer"). In the following, we present the code that integrated TC with the "apply"-style tactic, which can make the TC more powerful.
>
> * Recall the core idea of TC: replace potential incorrect tools/tactics with predefined correct ones. In this setting, we replace incorrect "apply"-style tactics with 12 predefined "apply"-style tactics.
> * Step 1:  Check whether we need to process the "apply" tactics. If we need and the previous tactic does not begin with "apply", then we directly change ‘apply'-style proofs to single-step method, which is sledgehammer+11 tactics. And if succeed via sledgehammer+11 tactics, then we continue. That is, if succeed via sledgehammer+11 tactics, for the next tactic, if it still begins with “apply”, we directly ignore it. If fails, we move to Step 2
> * Step 2: Try the given "apply"-style tactics. If fails, we move to Step 3.
> * Step 3: If Step 2 fails, then try sledgehammer+11 tactics. And if it succeeds, we continue. And if it fails, we try all 12 predefined "apply"-style tactics.
> * If Step 3 fails, then the prover fails.
>
>
>
>
> The following is the Python-like pseudocode.
>
> ```python
> #tactic_list: list of the tactics of formal proof
> #prover: Isabelle Prover
> #TCUsage: whether employ Tool Correction
> tool_heuristics=['by auto','by arith','by blast', 'by simp',
> 'by fastforce', 'by force', 'by eval', 'by presburger', 'by sos',
> 'by linarith', 'by (auto simp: field_simps)', 'sledgehammer']
>
>
> # can be other heuristics.
> apply_heuristics=['apply auto','apply arith','apply blast', 'apply simp',
> 'apply fastforce', 'apply force', 'apply eval', 'apply presburger', 'apply sos',
> 'apply linarith', 'apply (auto simp: field_simps)', 'apply assumption']
>
>
> output={}
> previous_tactics="None"
> for tactic in tactic_list:
>     use_heuristics=False
>
>     if tactic.strip().startswith("done") and output.has_key('ignore_apply'): #solve "apply"-style via "by".
>         previous_tactics="done"
>         continue
>
>     # Step 1 Begin
>     if tactic.strip().startswith("apply") and TCUsage:
>         if output.has_key('ignore_apply') and previous_tactics.startswith("apply"):
>             # solved by previous tactics, such as single-step
>             continue
>         elif not (previous_tactics.startswith("apply")) #If previous one is not "apply"
>             for tool_try in tool_heuristics: # try single-step
>                 output = prover.run_tac(tool_try)
>                 if output['error'] is None:
>                     break
>             if output['error'] is None:
>                 #go to next tactic, and ignore the following "apply"
>                 #becuase already finish the proof
>                 output['ignore_apply']=True #solve the proof
>                 continue
>         else:
>             #do nothing
>     # Step 1 End
>
>
>     previous_tactics=tactic
>
>     #Step 2 Begin
>     output = prover.run_tac(tactic)
>     #Step 2 End
>
>     if output['error'] is not None:
>         if TCUsage: # Use Tool Correction or Not
>             if tactic.strip().startswith("by") or tactic.strip()==("."):
>                 use_heuristic=True
>
>         if ("sledgehammer" in tactic) or use_heuristic:
>             for tool_try in tool_heuristics:
>                 output = prover.run_tac(tool_try)
>                 if output['error'] is None:
>                     output['ignore_apply']=True
>                     break
>
>
>         # Step 3 Begin
>         if TCUsage and tactic.strip().startswith("apply"):
>             for tool_try in tool_heuristics:
>                 output = prover.run_tac(tool_try)
>                 if output['error'] is None:
>                     output['ignore_apply']=True #solve the proof
>                     break
>
>             if output['error'] is not None:
>                 for apply_try in apply_heuristics:
>                     output = prover.run_tac(apply_try)
>                     if output['error'] is None:
>                         # if not use by-style tactic,
>                         # not sure whether the whole proof is solved or not here.
>                         # So that NO output['ignore_apply']=True
>                         break
>         # Step 3 End
>
>
>     if output['error'] is not None:
>         return "tactic_failed", output
>     if output['tactic_state'] == 'no goals':
>         return "success", output
>
> return "proof_incomplete", output
> ```
>
> If you think our comment and update address your concerns, could you please consider raising your rating of the paper? Thank you!

---

> > ### Author Response · Authors · 2023-11-23
> > **Waiting for further discussion**
> >
> > Dear reviewer Wc5t,
> >
> > Thanks again for your valuable time and insightful comments. As the deadline for the Author/Reviewer discussion is approaching, it would be nice of you to let us know whether our answers have solved your concerns so that we can better improve our work. We are looking forward to your feedback.
> >
> > Best regards, Paper 1096 Authors

---

### Author Response · Authors · 2023-11-14
**Author Response to All Reviewers (Part 1/2)**

Dear all reviewers:

We sincerely appreciate the reviewers for the time and efforts on the review. We first address some common questions, followed by detailed responses to each reviewer separately. We hope our responses clarify existing doubts. We will really appreciate it if Reviewer Wc5t and Reviewer NmsF can kindly reconsider the decision, provided that the main comments are well addressed.

**The difference between Tool Correction and sledgehammer+heuristic**

The difference between sledgehammer+heuristics and Tool Correction is the following.
* Different motivation.
   * DSP regards sledgehammer + heuristics as a more powerful proving tool, such as by simp, by fastforce and so on.
   * The motivation of Tool Correction comes from an observation: LLM formal proof can not prove a simple conjecture x = 19∗(x div 19)+4. After our analysis, this is caused by LLM hallucination that LLM wrong believes by (simp add: div mult mod eq) can prove x = 19∗(x div 19)+4. To mitigate the hallucination, we propose to post-process the LLM-generated formal proof by predefined rules, which is called Tool Correction. Following this direction,  the automated theorem proving performance can be further improved, if there are better rules to post-process the generated formal proof. And this is the one important point that we want to present and prove by Tool Correction.

* Different implementation.
   * The implementation of sledgehammer + heuristics (implemented by DSP): given a tactic, if the tactic is the sledgehammer, then utilize sledgehammer + 11 tactics to close the conjecture.
   * The implementation of Tool Correction:  1) given tactic, we try the given tactic (whatever it is); 2) If it fails and it begins with "by" or "." or “sledgehammer”, then turn to all potential tools, such as a sledgehammer, simp, and so on.  For example, if “by (simp add: div mult mod eq)” fails, the DSP will immediately stop validating, while our proposed Tool Correction will try all other potential tools.

* In this paper, to have a fair comparison with sledgehammer + heuristics, we employ the same tools, which are also sledgehammer + 11 tactics. To further improve the performance, more powerful proving tools can be easily added, such as by clarify, by fact, by rule, by erule, by elim, by induction, by algebra and so on.

* Insight of Tool Correction: 1) post-process of formal proof can significantly further improve performance; 2) future work could try to design a more powerful post-process method, such as post-processing a tactic beginning with “apply”.


We also provide the number of wrong steps fixed by Tool Correction, on the miniF2F-test dataset. The following is the number of passed tactics of sledgehammer+heuristics and Tool Correction
* Proof comes from: the miniF2F validation set (pass rate 55.3%) and test set result (pass rate 51.2%), with GPT-4, human informal proof, Conjecture Correction and Tool Correction.
* The definition of proof step: a proof step is regarded as a tactic.
* Calculation protocol: if sledgehammer+heuristic or Tool Correction fails to validate the current tactic, then the current proving process will be terminated and we will turn to the next formal proof validation. Finally, we calculate how many correct tactics/proof steps.
|  Dataset | Sledgehammer+heuristics   | Tool Correction  | Number of Fixed Steps|
|  ---- |  ----  | ----  | ----  |
| miniF2F-valid  |2260  | 3486 |1226 |
| miniF2F-test  |2594  | 3887 |1293 |

On miniF2F-valid, sledgehammer+heuristics can help the prover successfully pass 2260 steps. After adding Tool Correction, the number increases to 3486 steps. Therefore, Tool Correction fixes 1226 wrong steps. On the miniF2F-test, Tool Correction fixes 1293 wrong steps.

Moreover, we also compared the sledgehammer+heuristic and Tool Correction performance in Table 2, which is shown in the following. The DSP implementation contains the sledgehammer+heuristics implementation.
|  Method   | TC  | CC  | miniF2F-valid|miniF2F-test|
|  ----  | ----  | ----  | ----  | ----  |
| DSP  | &cross; |&cross; |50.4% |42.6% |
| DSP+TC  | &check; |&cross; |52.8% |45.9% |
| DSP+CC  | &cross; |&check; |46.7% | 43.0% |
| DSP+TC+CC (Lyra)  | &check; |&check; |55.3% |51.2% |

---

> ### Author Response · Authors · 2023-11-14
> **Author Response to All Reviewers (Part 2/2)**
>
> **The difference between Lyra and other works**
>
> The implementation difference between Lyra, Baldur [1], Coderl [2] and RustGen [3]  is the following.
>
> |  Method   | Training-Free  |Information Besides Response and Error  |Reset Initial solution| Control Indicator |
> |  ----  | ----  | ----  | ----  | ----  |
> | Baldur [1]  |&cross;  |&cross;  |&cross;  |&cross;  |
> | Coderl [2] |&cross;  |&cross;  |&cross;  |&cross;  |
> | RustGen [3]  |&check;  |&cross;  |&cross;  |&cross;  |
> | Lyra  | &check; |&check; |&check; |&check; |
>
> Baldur and Coderl need training, but Lyra is training-free. Compared with RustGen (designed for code generation) refining only with previous responses and error messages, the Lyra can sufficiently utilize problem information, informal proof information, previous response and the error message. Meanwhile, Lyra will reset the initial solution, if fails to refine the solution after several rounds. Also, Lyra utilizes “proof -” as the indicator to control the model response to be formal proof.
>
>
>
> Meanwhile, the performance comparison between Baldur [1], Thor [4], and Lyra is the following.
>
> |  Method   | AFP-Computer Science  | AFP-Logic  | AFP-Mathematics  | AFP-Tools|  miniF2F-valid|miniF2F-test|
> |  ----  | ----  | ----  | ----  | ----  | ----  | ----  |
> | Baldur [1]  | 50.0%  | 51.6%  | 41.9%  |53.9%  |-  |-  |
> | Thor [4]  |57.5% | 53.6% |  50.5%  |51.8%  |28.3%  |29.9%  |
> | Lyra  | - | - | -  |-  |55.3%  |51.2%  |
>
> As Baldur does not release code or conduct experiments on miniF2F, currently we cannot directly compare their performance. Fortunately, Baldur compared with Thor on AFP topic classification.
> * Compared to Baldur, on AFP topic classification, Thor achieves better performance in Computer Science, Logic and Mathematics. Also, they achieve comparable performance on Tool AFP topic classification. This suggests that Thor can achieve better performance than Baldur.
> * On the miniF2F dataset, our proposed Lyra is significantly better than Thor. Our Lyra achieves 55.3% on miniF2F-validation and 51.2% on miniF2F-test, while Thor achieves 28.3% on miniF2F-validation and 29.9% on miniF2F-test.
> * Therefore, Lyra should be a more powerful model than Baldur in the automated theorem proving area.
>
> [1] First, E., Rabe, M. N., Ringer, T., & Brun, Y. (2023). Baldur: whole-proof generation and repair with large language models. arXiv preprint arXiv:2303.04910.
>
> [2] Le, H., Wang, Y., Gotmare, A. D., Savarese, S., & Hoi, S. C. H. (2022). Coderl: Mastering code generation through pretrained models and deep reinforcement learning. Advances in Neural Information Processing Systems, 35, 21314-21328.
>
> [3] Wu, X., Cheriere, N., Zhang, C., & Narayanan, D. (2023). RustGen: An Augmentation Approach for Generating Compilable Rust Code with Large Language Models.
>
> [4] Jiang, A. Q., Li, W., Tworkowski, S., Czechowski, K., Odrzygóźdź, T., Miłoś, P., ... & Jamnik, M. (2022). Thor: Wielding hammers to integrate language models and automated theorem provers. Advances in Neural Information Processing Systems, 35, 8360-8373.

---

### Author Response · Authors · 2023-11-20
**Paper Update Summary**

Dear Reviewers and ACs:

We want to thank all reviewers for their time and energy. Their feedback has been hugely helpful and we truly appreciate it.

We have updated the paper since the paper was submitted. We summarize the major changes here. In the paper, the main updates are also highlighted in blue.
* Add the number of fixed wrong steps by Tool Correction.
* Discuss the relation between performance and the time limit.
* Explain “why the informal proof and the formal sketch are actually quite different”.
* Add a discussion paragraph about “why LLMs might be able to understand error messages”
* Various improvements in writing/references.

Thank you very much again!

Best regards, Paper 1096 Authors

---

### Author Response · Authors · 2023-11-23
**Summarize Contributions of the Paper**

Dear Reviewers and ACs,

Thank you all for your time and effort in reviewing this paper, and we are grateful to all the reviewers. The effectiveness of the proposed mechanism is endorsed by all reviewers. The experiment results are comprehensive and convincing, supported by reviewers 1Jjk, NmsF, and bwwN. The paper presentation and writing are well recognized by the reviewers Wc5t, 1Jjk and bwwN.

Additionally, we outline our paper's main contributions, including the additional conclusions during the rebuttal discussion phase:
* We propose Lyra, a novel approach that has both Tool Correction and Conjecture Correction, for automated theorem proving.
* The Tool Correction is significantly better than the sledgehammer+heuristics in DSP. Meanwhile, the proposed Tool Correction has good generalization ability. Moreover, the Tool Correction can be integrated with the "apply"-style tactic, while previous sledgehammer+heuristics can not. The creation of dynamic heuristics of TC can be an appealing direction, endorsed by the reviewer 1Jjk.
* The Conjecture Correction is a flexible approach for integrating error messages and models, with very few modifications. We propose two hypotheses to explain why the LLM can understand the error message.
* We conduct extensive experiments to prove and analyze the effectiveness of Tool Correction and Conjecture Correction.
* Lyra achieves the new state-of-the-art on miniF2F-valid (48.0% → 55.3%) and miniF2F-test (45.5% → 51.2%). And we solve 3 IMO problems that are difficult for previous models.

Best regards, Paper 1096 Authors

---

### Meta-Review · Area_Chair_u7DS · 2023-12-05

**Metareview:**

The paper received border line scores from the reviewers. The contributions presented are quite domain specific and not too novel. Some contributions seem to overlap with another paper that is submitted to ICLR this year. I have read the paper and I don't feel that it meets the ICLR bar for this year.

**Justification For Why Not Higher Score:**

Incremental improvement over previous work with domain specific techniques.

**Justification For Why Not Lower Score:**

N/A

---

### Decision · Program_Chairs · 2024-01-16

Reject